# Treatment Effect Estimation for Optimal Decision-Making

**Dennis Frauen**[*]
LMU Munich
Munich Center for Machine Learning

**Valentyn Melnychuk**
LMU Munich
Munich Center for Machine Learning

**Jonas Schweisthal**
LMU Munich
Munich Center for Machine Learning

**Mihaela van der Schaar**
University of Cambridge

**Stefan Feuerriegel**
LMU Munich
Munich Center for Machine Learning

## Abstract

Decision-making in various fields, such as medicine, is heavily based on conditional average treatment effects (CATEs). Practitioners commonly make decisions by checking whether the estimated CATE is positive, even though the decision-making performance of modern CATE estimators (meta-learners) is poorly understood. In this paper, we study optimal decision-making based on two-stage meta-learners (e.g., DR-learner), which estimate CATE via a second-stage regression. We show that these meta-learners can be suboptimal when used for decision-making in common settings where the second-stage regression is over a restricted function class (e.g., when using regularization or employing fairness/interpretability constraints). Intuitively, this occurs because such estimators prioritize CATE accuracy in regions far away from the decision boundary, which is ultimately irrelevant to decision-making. As a remedy, we propose a novel two-stage learning objective that re-targets the CATE to balance CATE estimation error and decision performance. We then propose a neural method that optimizes an adaptively-smoothed approximation of our learning objective. Finally, we confirm the effectiveness of our method both empirically and theoretically.

## 1 Introduction

Data-driven decision-making across various fields, such as medicine [19], public policy [1, 39], and marketing [58], relies on understanding how treatments affect different individuals and groups. This heterogeneity in the treatment effect across individuals is typically quantified through the *conditional average treatment effect (CATE)*. Then, a common approach from practice to obtain decisions from a CATE is **thresholding**: *individuals with positive CATE receive treatment, while those with negative CATE do not* [13]. For example, in medicine, clinicians typically administer treatments to the subset of patients who are expected to benefit from the intervention [37].

However, despite being widely used in practice, the optimality properties of such a thresholding approach for decision-making are unclear. We argue that minimizing estimation error and optimizing

---

[*]Correspondence to: frauen@lmu.de

39th Conference on Neural Information Processing Systems (NeurIPS 2025).

for decision-making performance are inherently different objectives. Existing literature acknowledges the distinction between CATE estimation error and decision-making performance [14] and draws connections between the two in specific situations [7, 13, 12]. Nevertheless, the optimality of decision-making based on modern CATE estimators remains unclear, and approaches are missing for how to improve thresholding-based decision rules.

In this paper, *we study optimal decision-making based on two-stage meta-learners*. Two-stage meta-learners estimate CATE by employing a second-stage regression over a prespecified function class. They are widely used in practice and include orthogonal learners such as the DR-learner [55, 33]. We show theoretically that, while these methods may be optimal for CATE estimation, *they can lead to suboptimal decisions when combined with a thresholding approach*, particularly when the second-stage model class is restricted. This is crucial in various real-world applications, in which the CATE model class is often restricted, e.g., due to fairness constraints [18, 35].

**Intuition:** *Why are two-stage learners suboptimal for decision-making?* Fig. 1 shows the results of one of our experimental setups (details in Sec. 5), where the ground-truth CATE (red line) is positive everywhere, except for a small region of the covariate space. An optimal policy is one that administers the treatment everywhere, except for that region (because the treatment is harmful in that region). Further, we show three two-stage learners with a regularized model class: the estimator in blue ($\gamma = 0$) achieves the lowest CATE estimation error, but yields the wrong policy in the region of negative CATE. In contrast, the estimators shown in violet (generated by our method that we propose later) is preferred for decision-making: it sacrifice a small amount of CATE accuracy to yield a better downstream decision performance, so that the decisions coincide with thresholding the ground-truth CATE.

To address the shortcomings of the thresholding approach from above, we propose *a novel second-stage learning objective that re-targets CATE to balance CATE estimation error and decision performance*. By doing so, we re-target the CATE to a new estimand which we call *policy-targeted CATE (PT-CATE)*. Our PT-CATE can still be approximately interpreted as a CATE, while leading to superior policies. We further propose a neural method to optimize our objective and estimate the corresponding PT-CATE. Our method follows a three-step procedure: first, we learn a neural network to estimate CATE; second, we learn a separate neural network to identify regions where incorrect decisions are made; and finally, we adjust the first neural network to improve decision-making in these problematic regions.

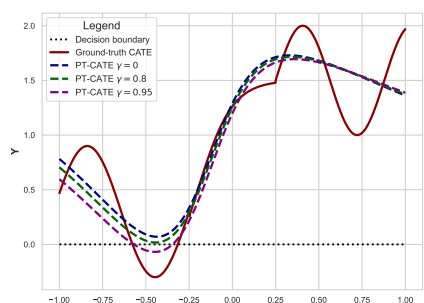

Figure 1: **Illustrative example showing the suboptimality of CATE estimation for decision-making.** The dotted lines show regularized two-stage CATE estimators. The blue line corresponds to standard two-stage CATE estimation, while the green and violet lines are generated by our method. The parameter $\gamma$ quantifies the trade-off between CATE estimation error and decision-making performance. Details are in Sec. 5.

Our **contributions**[2] are: (i) We show when two-stage CATE estimators with restricted model-classes can be suboptimal for decision-making. (ii) We develop a novel two-stage learning objective that effectively balances CATE accuracy and decision performance. For this, introduce a new estimand for policy-targeted CATE estimation (PT-CATE). (iii) We propose a neural method for our learning objective with theoretical guarantees and empirically demonstrate its ability to effectively trade-off CATE estimation error and decision-making performance.

## 2 Related work

Our work connects to several literature streams below (see Appendix A for an extended version).

**CATE estimation.** Methods for CATE estimation can be broadly categorized into (i) *model-based* and (ii) *model-agnostic* approaches. (i) Model-based methods propose specific models, such as regression forests [21, 59] or tailored neural networks [23, 50] for CATE estimation. (ii) Model-

---

[2]Code is available at `https://github.com/DennisFrauen/CATEForPolicy`.

agnostic methods (also called *meta-learners*) are "recipes" for constructing CATE estimators that can be combined with arbitrary machine learning models (e.g., neural networks) [38, 9].

Meta-learners often follow a two-stage approach to account for the fact that the CATE is often structurally simpler than its components (response functions) [33]. Prominent examples of two-stage meta-learners include the RA/X-learner [38], IPW-learner [9], and the DR-learner [55, 33]. The DR-learner has the additional advantage of being Neyman-orthogonal and thus being provably robust against estimation errors from the first-stage regression [8, 16]. In this paper, we show that such two-stage learners can be suboptimal for decision-making when using restricted second-stage model classes and propose a method to improve them.

**(Direct) Off-policy learning (OPL).** The goal in OPL is to directly learn an optimal policy by maximizing the so-called *policy value*. Approaches for estimating the policy value from data follow three primary approaches: (i) the direct method (DM) [47] leverages estimates of the response functions; (ii) inverse propensity weighting (IPW) [51] re-weights the data such that they resemble samples under the evaluation policy; and (iii) the doubly robust method (DR) [11, 2] combines both. Recent work has focused on enhancing finite-sample performance through techniques such as reweighting [24, 25] and targeted maximum likelihood estimation [5]. Further, extended versions have been developed for specific scenarios, including distributional robustness [31], fairness considerations [18], and continuous treatments [30, 49].

OPL is different from our work as follows: OPL aims to *directly learn* a policy, thus bypassing the need to estimate a CATE for decision-making. While this approach can optimize decision-making performance, it often does so at the expense of making black-box decisions that are not based on treatment effects. In contrast, our work prioritizes the interpretability inherent in CATE-based methods while leveraging insights from OPL to improve decision-making performance. This distinction is particularly relevant in fields like medicine, where the effectiveness of treatments is often evaluated using CATEs, and the CATEs are then used to guide interpretable clinical decisions [15].

**CATE estimation vs. decision-making.** In practice, it is common to use CATE estimators for decision-making through thresholding [e.g., 13, 37]. Yet, few works have formally studied the effectiveness of this approach. One literature stream discusses the suboptimality of estimating CATE for decision-making as compared with outcome/ response function modeling [14, 13, 12]. In this context, Zou et al. [63] study a continuous treatment setting for counterfactual prediction and propose to re-weight the loss with the inverse magnitude of the treatment effect. Additional works propose to adjust the objective to tailor the learning process towards decision-making [4, 52]. However, these works do not focus two-stage meta-learners under model class constraints.

Relatedly, Bonvini et al. [7] establish minimax-optimality results on the OPL performance of thresholded two-stage meta-learners, but only under certain assumptions (e.g., assuming that the second-stage model is well-specified). In contrast, we allow for the misspecification of second-stage models (e.g., by incorporating fairness or interpretability constraints) and instead show in such cases that two-stage learners may be suboptimal. Finally, [32] considers learning optimal treatment effect *rankings* under possible resource constraints while we consider thresholding, i.e., treating everyone that benefits from treatment.

**Research gap:** To the best of our knowledge, we are the first to show the suboptimality of two-stage meta-learners for decision-making under model class restrictions along with proposing novel methods for improving decision performance. Our work thus bridges a critical gap between the theoretical understanding and the practical application of CATE estimation for decision-making.

# 3 Problem setup

## 3.1 Setting

**Data:** We consider a standard causal inference setting with a population $Z = (X, A, Y) \sim \mathbb{P}$, where $X \in \mathcal{X} \subseteq \mathbb{R}^d$ are observed pre-treatment covariates, $A \in \{0, 1\}$ is a binary treatment (or action), and $Y \in \mathbb{R}$ is an outcome (or reward) of interest that is observed after the treatment $A$. We assume that we have access to a dataset (either randomized or observational) $\mathcal{D} = \{(x_i, a_i, y_i)\}_{i=1}^n$ of size $n \in \mathbb{N}$ sampled i.i.d. from $\mathbb{P}$. For example, in a medical setting, $X$ are patient covariates, $A$ is a medical

drug, and $Y$ is a health outcome (e.g., blood pressure). Another example is logged data of A/B tests in marketing, where $X$ are user demographics, $A$ is a binary decision of whether a coupon was given, and $Y$ is some reward such as user engagement.

**Notation.** We define the *response functions* as $\mu_a(x) = \mathbb{E}[Y \mid X = x, A = a]$ for $a \in \{0, 1\}$ and the *propensity score* (behavioral policy) as $\pi_b(x) = \mathbb{P}(A = 1 \mid X = x)$. We refer to these functions as *nuisance functions*, denoted by $\eta = (\mu_1, \mu_0, \pi_b)$. A *policy* is any function $\pi\colon \mathcal{X} \to [0, 1]$ that maps an individual with covariates $X \in \mathcal{X}$ to a probability $\pi(X)$ of receiving treatment.

**Identifiability:** We use the potential outcomes framework [48] and denote $Y(a)$ as the potential outcome corresponding to a treatment intervention $A = a$. The potential outcomes are not directly observed, which means that we have to impose assumptions to identify any estimands from data.

**Assumption 3.1** (Standard causal inference assumptions). *For all $a \in \{0, 1\}$ and $x \in \mathcal{X}$ it holds:*
(i) *consistency*: $Y(a) = Y$ whenever $A = a$; (ii) *overlap*: $0 < \pi_b(x) < 1$ whenever $\mathbb{P}(X = x) > 0$; *and* (iii) *ignorability*: $A \perp Y(1), Y(0) \mid X = x$.

Assumption 3.1 is standard in the causal inference literature [56, 9]. (i) Consistency prohibits interference between individuals; (ii) overlap ensures that both treatments are observed for each covariate value; and (iii) ignorability excludes unobserved confounders that affect both the treatment $A$ and the outcome $Y$. Note that (ii) and (iii) are usually fulfilled in randomized experiments, which fall within our setting.

## 3.2 Mathematical preliminaries

**Policy value.** The decision-making performance of a policy $\pi$ is usually quantified via its *policy value*. Formally, the policy value is defined via

$$V(\pi) = \mathbb{E}[Y(\pi(X))] = \mathbb{E}[\pi(X)Y(1) + (1 - \pi(X))Y(0)] = \mathbb{E}[\pi(X)\mu_1(X) + (1 - \pi(X))\mu_0(X)]. \tag{1}$$

Under Assumption 3.1, it is identified via $V(\pi) = \mathbb{E}[\pi(X)\mu_1(X) + (1 - \pi(X))\mu_0(X)]$, and thus can be estimated from the available data.

**CATE.** We define the *conditional average treatment effect* (CATE) as

$$\tau(x) = \mathbb{E}(Y(1) - Y(0) \mid X = x] = \mu_1(x) - \mu_0(x), \tag{2}$$

where identifiability in terms of response functions $\mu_a(x)$ follows again from Assumption 3.1. The CATE captures heterogeneity in the treatment effect across individuals characterized by $X$.

**CATE estimation.** A straightforward approach to estimating the CATE is the so-called plug-in approach. Here, we first obtain estimators for the response functions $\hat{\mu}_1$ and $\hat{\mu}_0$ (which are standard regression tasks) and then obtain a CATE estimator via $\hat{\tau}_{\mathrm{PI}}(x) = \hat{\mu}_1(x) - \hat{\mu}_0(x)$.

However, it is well known that the plug-in approach is suboptimal as it suffers from so-called plug-in bias [34]. In contrast, state-of-the-art approaches for CATE estimation are based on the following two-stage principle: in stage 1, one obtains estimators $\hat{\eta}$ of the nuisance functions $\eta$, and, in stage 2, one estimates the CATE directly via a second-stage regression

$$\hat{\tau}_{\mathcal{G}} = \arg\min_{g \in \mathcal{G}} \mathcal{L}_{\hat{\eta}}(g), \tag{3}$$

where $\mathcal{G}$ is some function class and $\mathcal{L}_\eta(g)$ is a (population) second-stage loss.

Two-stage CATE meta-learners offer two key advantages: First, they enable to direct incorporationcof onstraints on the CATE estimate, such as fairness requirements [45, 35] or interpretability conditions [54]. Second, it leverages the inductive bias that the CATE structure is typically simpler than its constituent response functions, making direct estimation more effective [33].

**Direct OPL**: One approach for obtaining an optimal policy $\pi^*$ is direct off-policy learning (OPL): here, we directly maximize the estimated policy value and solve $\pi^*_{\mathrm{OPL}} = \arg\max_{\pi \in \Pi} \hat{V}(\pi)$ for some estimator $\hat{V}(\pi)$ of $V(\pi)$ over a prespecified class of policies $\Pi$. An advantage of the OPL approach is that it directly optimizes for decision-making performance. However, the obtained $\pi^*$ is a black-box policy and may be hard to provide with meaningful interpretation.

**CATE-based OPL.** Another common approach, which we focus on in this paper, is to use the CATE $\tau(x)$ for decision-making by *thresholding* [13]. The approach has two steps. First, the CATE

$\hat{\tau}$ is estimated via e.g., a two-stage meta-learner. Second, the CATE-based policy is obtained via $\pi_{\hat{\tau}}(x) = \mathbf{1}(\hat{\tau}(x) > 0)$. The treatment is thus only applied to individuals with a positive CATE (= individuals for which the treatment helps on average). To see why this is a valid approach note that we can write the policy value as

$$V(\pi) = \mathbb{E}[\pi(X)(\mu_1(X) - \mu_0(X)) + \mu_0(X)] \propto \mathbb{E}[\pi(X)\tau(X)], \qquad (4)$$

where $\propto$ denotes equivalence up to a constant (irrelevant to maximization). Hence, the (ground-truth) thresholded CATE policy $\pi_{\tau}(x)$ maximizes the policy value if $\pi_{\tau} \in \Pi$.

A benefit of CATE-based policy learning is that the CATE provides individualized estimates of the incremental benefits from treatment. Unlike direct OPL methods, which yield black-box policies optimized solely for overall performance, the CATE $\tau(X)$ explicitly allows to quantify the net gain from treatment. As a result, CATE-based policy allows to compare the estimated treatment effects against domain knowledge. Further, CATE-based policy learning enables practitioners to weigh the benefits against potential side effects when making treatment decisions, a critical consideration in domains like personalized medicine.

### 3.3 Research questions

In this paper, we study the optimality of CATE-based policy learning when the CATE estimator $\hat{\tau}_{\mathcal{G}}$ is obtained via a second-stage regression over a function class $\mathcal{G}$ (as in Eq. (3)). More formally:

> ① *Do two-stage estimators $\hat{\tau}_{\mathcal{G}}$ yield policies $\pi_{\hat{\tau}}$ that maximize the policy value $V(\pi)$ among thresholded policies $\pi \in \Pi_{\mathcal{G}} = \{\mathbf{1}(g > 0) \mid g \in \mathcal{G}\}$?*

If $\tau \in \mathcal{G}$ (i.e., $\mathcal{G}$ contains the ground-truth CATE), optimality (in population) of $\pi_{\hat{\tau}}$ is guaranteed by Eq. (4). However, in two-stage CATE estimation, $\mathcal{G}$ is often restricted such that $\tau \notin \mathcal{G}$. This occurs, for instance, when fairness or interpretability constraints are imposed [54, 35], or when regularization is applied to smooth the second-stage model [33]. In this setting, we later show in Sec. 4.1 that there can exist policies $\pi \in \Pi_{\mathcal{G}}$ with $V(\pi) > V(\pi_{\hat{\tau}})$. In other words, thresholding a two-stage CATE estimator may *not* yield an optimal policy, *even* when the policy class is restricted in an analogous manner. This leads to our second research question, where we seek a policy that achieves (i) a low CATE estimation error and (ii) a good decision performance:

> ② *How can we learn a function $g \in \mathcal{G}$ that satisfies two key properties*: (i) $g \approx \tau$ ($g$ is a good approximation of the CATE), and (ii) $\pi_g(x) = \mathbf{1}(g(x) > 0)$ is approximately optimal, that is, $V(\pi_g) \approx V(\pi_{\mathcal{G}}^*)$, where $\pi_{\mathcal{G}}^*$ is an optimal policy among the class $\Pi_{\mathcal{G}}$?

## 4 Re-targeting CATE for decision-making

We now answer both research questions from Sec. 3.3. First, in Sec. 4.1, we show the suboptimality of two-stage CATE estimators for decision-making when $\tau \notin \mathcal{G}$. Then, in Sec. 4.2, we propose a new learning objective that balances CATE estimation error and policy value. Finally, in Sec. 4.3 and Sec. 4.4, we propose a two-stage learning algorithm and provide theoretical guarantees.

### 4.1 Suboptimality of CATE for decision-making

To provide an intuition on why two-stage CATE estimators can be suboptimal for decision-making, we first consider a toy example illustrated in Fig. 2 (left). Here, we examine a two-stage CATE estimator with one-dimensional covariates $X$ and $\mathcal{G} = \{g(x) = ax + b \mid a, b \in \mathbb{R}\}$ being the class of linear functions. Hence, the policy class we consider is $\Pi_{\mathcal{G}} = \{\mathbf{1}(ax + b > 0)\}$, which represents the class of thresholded linear policies. The ground-truth CATE is nonlinear so that $\tau \notin \mathcal{G}$.

We make two key observations: (i) The optimal policy $\pi^* = \arg\max_{\pi \in \Pi_{\mathcal{G}}} V(\pi)$ assigns a treatment in the region of the covariate space where ground-truth CATE is positive, but no treatment in the region where it is negative. This is equivalent to thresholding the ground-truth CATE (represented by the red line). (ii) The optimal linear approximation to the CATE is $g^* \in \arg\min_{g \in \mathcal{G}} \mathbb{E}[(\tau(X) - g(X))^2]$

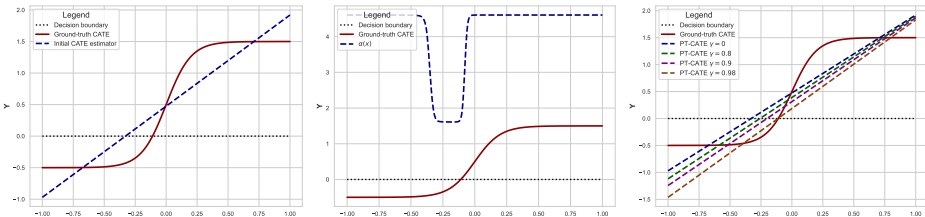

Figure 2: **Experimental results for our proposed method with $\mathcal{G}$ being the class of linear models.** *Left:* CATE estimator (blue) is the best linear approximation of the (nonlinear) ground-truth CATE (red). *Center:* the trained $\alpha(X)$ detects the region in which the estimated CATE has the wrong sign. *Right:* re-targeted CATE estimators using our proposed loss with trained $\alpha(X)$ and different $\gamma$ values.

(represented by the blue line). Note that the blue line does not intersect the $x$-axis at the same point as the true CATE. As a result, there exists a region where the policy $\pi_{g^*}(x) = \mathbf{1}(g^*(x) > 0)$ makes the wrong treatment decision. Thus, the policy $\pi_{g^*}$ is suboptimal, i.e., $V(\pi^*) > V(\pi_{g^*})$. The optimal policy is instead obtained by thresholding a linear function that intersects the $x$-axis at the same point as the ground-truth CATE.

To generalize the above example, we derive the following theorem. We denote $V_\tau(\pi) = \mathbb{E}[\tau(X)\pi(X)]$ to note the dependency of the policy value on the underlying ground-truth $\tau$.

**Theorem 4.1** (Suboptimality of CATE-based decision-making). *Let $\mathcal{G}$ be a class of neural networks with fixed architecture. Then, there exists a CATE $\tau^* \notin \mathcal{G}$, so that, for any optimal CATE approximation $g^*_{\tau^*} \in \arg\min_{g \in \mathcal{G}} \mathbb{E}[(\tau^*(X) - g(X))^2]$, it holds that $V_{\tau^*}(\pi_{g^*_{\tau^*}}) < V_{\tau^*}(\pi^*_{\tau^*})$, for any optimal policy $\pi^*_{\tau^*} \in \arg\max_{\pi \in \Pi_\mathcal{G}} V_{\tau^*}(\pi)$.*

*Proof.* See Appendix B. □

**Interpretation.** Theorem 4.1 demonstrates that, regardless of how we choose our class $\mathcal{G}$ in the second-stage CATE regression, there always exists a ground-truth CATE for which the estimated CATE is suboptimal in terms of thresholded decision-making ($\rightarrow$ thus answering ①). Specifically, there always exists a more optimal function $g \in \mathcal{G}$ which, while not necessarily the best CATE approximation, yields improved policy value.

In general, the discrepancy between CATE-based and optimal policy value is determined by how well the CATE projection using the model class $\mathcal{G}$ can estimate the sign of the ground-truth CATE. This depends on several factors, including (i) the error of the optimal projection (related to model complexity), and (ii) the structure of the ground-truth CATE. As an illustration, consider Figure 2 (left), where the discrepancy in policy value is characterized by the distance of the two points where the blue (estimator) and red line (ground-truth CATE) intersect zero. For (i), we could make the red line steeper (increasing the projection error), and thus can arbitrarily widen this gap to obtain an arbitrary discrepancy between CATE-based and optimal policy value. For (ii), we could simply shift the red line upwards, which would also widen the gap without changing the projection error. In Appendix B, we present an additional result bounding the discrepancy in policy values if we can bound the CATE projection error via $\mathcal{G}$.

## 4.2 A novel learning objective for re-targeting CATE

**Basic idea.** Motivated by our previous analysis, we propose a learning objective that learns a re-targeted CATE, leading to an improved policy value while maintaining interpretability as an approximate CATE estimator ($\rightarrow$ thus answering ②). Our motivation comes from Fig. 2 (right), in which we observe that there exists a continuous set of solutions between the optimal CATE approximation and the optimal linear function that maximizes policy value (after thresholding). The basic idea is to minimize a convex combination of the CATE estimation error and the negative policy value of the thresholded policy.

**Definition 4.2.** *We define the $\gamma$-**policy-targeted CATE** ($\gamma$-PT-CATE) as any solution $g^*$ minimizing*

$$\mathcal{L}_\gamma(g) = (1 - \gamma)\mathbb{E}[(\tau(X) - g(X))^2] - \gamma\mathbb{E}[\mathbf{1}(g(X) > 0)\tau(X)]. \tag{5}$$

*over a class of function $g \in \mathcal{G}$.*

The hyperparameter $\gamma$ controls the trade-off between CATE accuracy and policy value optimization. For $\gamma = 0$, the objective reduces to standard CATE estimation, while, for $\gamma = 1$, corresponds to pure policy value maximization (OPL) and thus disregards the CATE estimation error. We discuss principled methods for selecting $\gamma$ in Section 4.4.

**Optimization challenges.** The loss in Eq. (5) does not allow for gradient-based optimization due to the non-differentiability of the indicator function $\mathbf{1}(g(X) > 0)$. A naïve approach would be to use a smooth approximation of the indicator via the sigmoid function $\sigma(\alpha g(X))$ for a sufficiently large $\alpha$. However, this introduces a challenging trade-off: large values for $\alpha$ provide a better approximation to the indicator function but suffer from vanishing gradients, while small values for $\alpha$ maintain useful gradients but poorly approximate the indicator function. Furthermore, small values for $\alpha$ may incentivize the model to compensate by increasing $g(X)$, thereby degrading CATE quality.

**Adaptive indicator approximation.** We address this optimization challenge by introducing $\alpha(X) > 0$ as a function of the covariates $X$. That is, we approximate $\mathcal{L}_\gamma(g)$ from Eq. (5) via

$$\mathcal{L}_{\gamma,\alpha}(g) = (1 - \gamma)\,\mathbb{E}[(\tau(X) - g(X))^2] - \gamma\,\mathbb{E}\Big[\tau(X)\sigma\left(\alpha(X)g(X)\right)\Big], \tag{6}$$

for some fixed adaptive approximation $\alpha(X) > 0$. Such an adaptive approach allows $\alpha$ to be large when the sign of $g$ is correct, thereby providing an improved indicator approximation in regions where no signal from the gradient from the policy value term is needed. Fig. 2 illustrates this concept, where we show an estimated CATE that is suboptimal for decision-making (left plot). The $\alpha(X)$ in Fig. 2 (center) is effective in identifying the region in the covariate space where the sign of $g$ is incorrect and provides gradients for these regions. Once we obtain a suitable $\alpha(X)$, we can minimize $\mathcal{L}_\gamma(g)$ to re-target the CATE estimate in regions of suboptimal decision-making (right plot).

**Learning $\alpha(X)$.** We obtain a loss for learning $\alpha(X)$ for fixed $g$ by transforming the OPL component in Eq. (5) into a classification problem (following an approach similar to, e.g., [61, 3]). We can write

$$V(\pi_g) \propto \mathbb{E}[\tau(X)\pi_g(X)] = \mathbb{E}[|\tau(X)|\,\pi_g(X)\,\mathrm{sgn}(\tau(X))] \tag{7}$$

By noting that maximizing $\mathbf{1}(g(X) > 0)\,\mathrm{sgn}(\tau(X))$ is equivalent to minimizing the binary cross-entropy loss over $g$ with label $\mathbf{1}(\tau(X) > 0)$, we can obtain $\alpha$ for fixed $g$ by minimizing

$$\mathcal{L}_{\gamma,g}(\alpha) = \mathbb{E}\Big[\,\big|\tau(X)\big|\,\ell\big(\alpha(X)\,g(X);\,\tau(X)\big)\Big], \tag{8}$$

where $\ell(u; y) = -\mathbf{1}(y > 0)\log(\sigma(u)) - \mathbf{1}(y < 0)\log(1 - \sigma(u))$, subject to $\alpha(x) \in [a, \infty)$ for all $x \in \mathcal{X}$ and $0 < a$. The scalar $a$ can be tuned by minimizing the loss from Eq. (5) on a validation set.

**Interpretation as stochastic policy.** The policy $\pi_{\alpha,g}(x) = \sigma(\alpha(x)g(x))$ can be interpreted as the best stochastic policy that is achievable for a fixed $g \in \mathcal{G}$. Here, the CATE approximation $g$ determines the sign (i.e., whether to give treatment or not), while the approximation $\alpha$ determines the stochasticity of the resulting policy. As shown in Fig. 2, $\alpha(x)$ will be large whenever $g(x)$ has the correct sign, therefore providing a policy $\pi_{\alpha,g}(x)$ that is closer to being deterministic.

### 4.3 Estimated nuisance functions

So far, we have assumed that the true CATE $\tau(X)$ is known, which is not the case in practice. To address this, we employ a *two-stage* estimation procedure similar to established CATE estimators [9, 33]. In the *first stage*, we obtain estimators $\hat{\eta}$ of the nuisance functions, $\eta = (\mu_1, \mu_0, \pi)$. These are standard regression or classification tasks that can be solved using various model-based methods from the literature [50, 59]. In the *second-stage*, we substitute these first-stage estimates into a second-stage loss that coincides with Eq. (8) and Eq. (6) in expectation.

To start with, we define

$$\mathcal{L}^m_{\gamma,\alpha,\eta}(g) = (1-\gamma)\,\mathbb{E}[(Y^m_\eta - g(X))^2] - \gamma\,\mathbb{E}\Big[Y^m_\eta\,\sigma\left(\alpha(X)g(X)\right)\Big] \text{ and } \mathcal{L}^m_{\gamma,g,\eta}(\alpha) = \mathbb{E}\Big[\,|Y^m_\eta|\,\ell(\alpha(X)\,g(X);\,Y^m_\eta)\Big], \tag{9}$$

where $Y^m_\eta$ is one of the following pseudo-outcomes:

$$
\begin{aligned}
Y^{\mathrm{PI}}_\eta &= \mu_1(X) - \mu_0(X), & Y^{\mathrm{RA}}_\eta &= A\,(Y - \mu_0(X)) + (1 - A)\,(\mu_1(X) - Y), \\
Y^{\mathrm{IPW}}_\eta &= \frac{(A - \pi_b(X))\,Y}{\pi_b(X)\,(1 - \pi_b(X))}, & Y^{\mathrm{DR}}_\eta &= \mu_1(X) - \mu_0(X) + \frac{(A - \pi_b(X))\,(Y - \mu_A(X))}{\pi_b(X)\,(1 - \pi_b(X))}.
\end{aligned}
\tag{10}
$$

**Theoretical analysis.** We now justify our pseudo-outcome-based loss from Eq. (9) theoretically. The first result shows that minimizing $\mathcal{L}^m_{\gamma,\alpha,\eta}(g)$ provides a meaningful minimizer.

**Theorem 4.3** (Consistency). *If the nuisance functions are perfectly estimated (i.e., $\hat{\eta} = \eta$), the pseudo-outcome loss $\mathcal{L}_{\gamma,\alpha,\eta}^{m}(g)$ has the same minimizer as $\mathcal{L}_{\gamma,\alpha}^{m}(g)$ w.r.t. $g \in \mathcal{G}$ for all $\alpha$ and $m \in \{\mathrm{PI}, \mathrm{RA}, \mathrm{IPW}, \mathrm{DR}\}$.*

*Proof.* See Appendix B. □

In practice, we use estimated nuisance functions $\hat{\eta}$, which means that Theorem 4.3 may not hold for $\mathcal{L}_{\gamma,\alpha,\hat{\eta}}^{m}(g)$ due to possible nuisance estimation errors. However, the following results provides an upper bound on how much the minimizer can deviate in the presence of estimation errors.

**Theorem 4.4** (Error rates). *Let $g^{*} = \arg\min_{g \in \mathcal{G}} \mathcal{L}_{\gamma,\alpha,\eta}^{m}(g)$ and $\hat{g} = \arg\min_{g \in \mathcal{G}} \mathcal{L}_{\gamma,\alpha,\hat{\eta}}^{m}(g)$ be the minimizers of the PT-CATE loss with ground-truth and estimated nuisances. Then, under the additional assumptions listed in Appendix B, it holds*

$$||g^{*} - \hat{g}||^{2} \lesssim R_{\gamma,\alpha,\hat{\eta}}^{m}(\hat{g}, g^{*}) + M_{\hat{\eta},\eta}^{m}\left((1-\gamma) + \gamma C_{\alpha}\right), \tag{11}$$

*where $|| \cdot ||$ is the $L^{2}$-norm, $R_{\gamma,\alpha,\hat{\eta}}^{m}(\hat{g}, g^{*}) = \mathcal{L}_{\gamma,\alpha,\hat{\eta}}^{m}(\hat{g}) - \mathcal{L}_{\gamma,\alpha,\hat{\eta}}^{m}(g^{*})$ is an optimization-dependent term, $C_{\alpha} > 0$ is a constant depending on $\alpha$, and $M_{\hat{\eta},\eta}^{m}$ is the (pseudo-outcome-dependent) rate term, defined via*

$$M_{\hat{\eta},\eta}^{\mathrm{PI}} = M_{\hat{\eta},\eta}^{\mathrm{RA}} \propto ||\hat{\mu}_{1} - \mu_{1}||^{2} + ||\hat{\mu}_{0} - \mu_{0}||^{2}, \quad M_{\hat{\eta},\eta}^{\mathrm{IPW}} \propto ||\hat{\pi}_{b} - \pi_{b}||^{2}, \tag{12}$$

$$M_{\hat{\eta},\eta}^{\mathrm{DR}} \propto ||\hat{\pi}_{b} - \pi_{b}||^{2}\left(||\hat{\mu}_{1} - \mu_{1}||^{2} + ||\hat{\mu}_{0} - \mu_{0}||^{2}\right). \tag{13}$$

*Proof.* See Appendix B. □

Theorem 4.4 shows that, as long as we are able to estimate the nuisance functions $\eta$ involved in the corresponding pseudo-outcome reasonably well, we can ensure a sufficiently good second-stage learner. Importantly, the doubly robust pseudo-outcome leads to a *doubly robust nuisance error rate*: only *either* the propensity score $\pi_{b}$ *or* the response functions $\mu_{a}$ need to be estimated well for the second-stage learner to converge well.

### 4.4 Learning algorithm

We provide a concrete learning algorithm to obtain $g(x)$ and $\alpha(x)$ from finite data. Given a dataset $\mathcal{D} = \{(x_{i}, a_{i}, y_{i})\}_{i=1}^{n}$, we can define empirical versions $\hat{\mathcal{L}}_{\gamma,\alpha,\hat{\eta}}^{m}(g)$ of $\mathcal{L}_{\gamma,\alpha,\hat{\eta}}^{m}(g)$ and $\hat{\mathcal{L}}_{\gamma,g,\hat{\eta}}^{m}(\alpha)$ of $\mathcal{L}_{\gamma,g,\hat{\eta}}^{m}(\alpha)$ by replacing expectations with empirical means. To minimize both empirical losses, we propose to parametrize $\alpha_{\phi}$ and $g_{\theta}$ as neural networks, where $\phi$ and $\theta$ denote their respective parameters. For training, we propose a three-step iterative learning algorithm (shown in Fig. 3):

• **Step 1 (initial CATE estimation):** train $g_{\theta}$ by minimizing $\hat{\mathcal{L}}_{\gamma=0,\alpha_{\phi},\hat{\eta}}^{m}(g)$ over $\theta$, using randomly initialized $\alpha_{\phi}$ with $\phi$ frozen. This gives an initial CATE estimator. • **Step 2 (region detection):** train $\alpha_{\phi}$ by minimizing $\hat{\mathcal{L}}_{\gamma,g_{\theta},\hat{\eta}}^{m}(\alpha_{\phi})$ over $\phi$, keeping $\theta$ frozen. The objective is for $\alpha_{\phi}$ to identify covariate regions where $g_{\theta}$ produces incorrect predictions of the sign. • **Step 3 (CATE refinement):** Retrain $g_{\theta}$ by minimizing $\hat{\mathcal{L}}_{\gamma,\alpha_{\phi},\hat{\eta}}^{m}(g)$ over $\theta$, with $\phi$ frozen. This step corrects $g_{\theta}$ in regions previously identified as having incorrect sign predictions. Fig. 2 shows experimental results for each of the three steps of our algorithm. Steps 2 and 3 can be repeated iteratively until convergence. The pseudocode is in Appendix C.

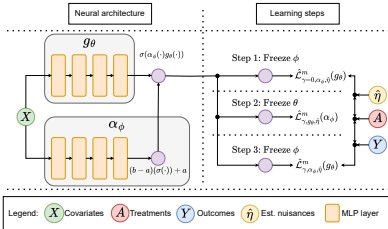

Figure 3: **Overview** of our second-stage architecture and our learning algorithm.

**Selecting $\gamma$.** The parameter $\gamma$ quantifies the trade-off between CATE estimation error and decision performance. Setting $\gamma = 0$ corresponds to standard CATE estimation, while $\gamma = 1$ corresponds to OPL (ignoring a meaningful CATE estimation completely). As such, the selection of $\gamma$ is mainly driven by domain knowledge. In practice, we recommend plotting the estimation CATE error and policy value (e.g., as in Fig. 4) and choosing $\gamma$ accordingly, or comparing trained models for multiple values of $\gamma$ (sensitivity analysis). If the main objective is optimal decision-making, practitioners may

choose $\gamma \approx 1$. The corresponding PT-CATE can be interpreted as the *best CATE approximation among all functions that can be thresholded for optimal decision-making* as long as $\gamma < 1$.

Additionally, the choice of $\gamma$ should reflect the expected misspecification of the function class $\mathcal{G}$. If $\mathcal{G}$ is overly flexible, then $\gamma \approx 0$ may already yield near-optimal decisions, as $\mathcal{G}$ is capable of closely approximating the true CATE. However, in many real-world scenarios $\mathcal{G}$ is deliberately constrained, e.g., for interpretability, fairness, or due to limited sample sizes. In such cases, selecting a larger $\gamma$ allows to compensate for this model misspecification.

## 5 Experiments

We now confirm the effectiveness of our proposed learning algorithm empirically. As is standard in causal inference [50, 9, 33], we use data where we have access to ground-truth values of causal quantities. We also provide experimental results using real-world data. Additional experimental results and robustness checks are reported in Appendix F.

**Implementation details.** We use standard feed-forward neural networks with tanh activations for $g_\theta$ and with ReLU activations for $\alpha_\phi$. We use $\rho(x) + a$ as the final activation function for $\alpha_\phi$ to ensure $\alpha_\phi(x) > a$, where $\rho(x)$ denotes the softplus function. We perform training using the Adam optimizer [36]. Further details regarding architecture, training, and hyperparameters are in Appendix C.

**Evaluation.** We evaluate a function $g$ learned by our method using two established metrics [50, 24]: (i) the *precision of estimating heterogeneous treatment effects (PEHE)* $\hat{\mathbb{E}}_n[(g(X) - \tau(X))^2]$, which quantifies the CATE estimation error, and (ii) the *policy loss* (negative policy value) given by $-\hat{\mathbb{E}}_n[\mathbf{1}(g(X) > 0)\tau(X)]$. For the experiments using simulated datasets, we use the known ground-truth CATE $\tau(X)$ for evaluation. For the experiments using real-world data, we evaluate by using the doubly robust pseudo-outcome $Y_{\hat{\eta}}^{\mathrm{DR}}$ instead, as the ground-truth CATE is not available.

**Baselines.** Standard two-stage CATE learners (i.e., PI/ RA/ IPW/ DR-learner) correspond to our method when setting $\gamma = 0$. To ensure a fair comparison, we use the *same* neural network architecture for all values of $\gamma$ in our experiment. We refrain from benchmarking with specific model architectures as we are not claiming general state-of-the-art results using our specific implementation. Additional experiments using different model architectures are in Appendix F. Of note, OPL methods can be viewed as a special case of our method when setting $\gamma = 1$.

*Simulated data.* **Experiments with ground-truth nuisance functions.** Fig. 1 and Fig. 2 (right) show the results of stage 2 of our PT-CATE algorithm for different values of $\gamma$ when using ground-truth nuisance functions in the first stage of Algorithm 1. The results show visually that our algorithm is effective in improving the decision threshold (and, thus, the policy value) as compared to the result for $\gamma = 0$, while maintaining good CATE approximations.

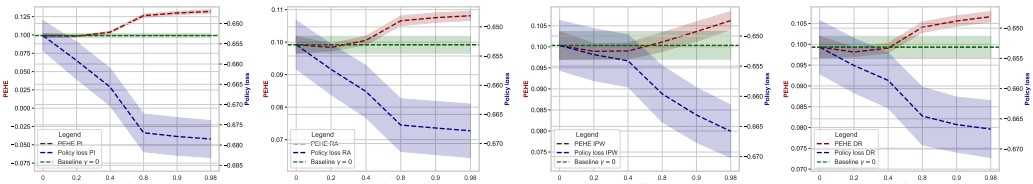

Figure 4: **Experimental results for setting A.** Shown: PEHE and policy loss over $\gamma$ (lower = better). Shown: mean and standard errors over 5 runs.

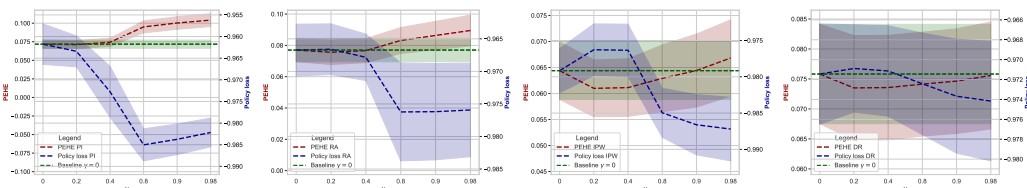

Figure 5: **Experimental results for setting B.** Shown: PEHE and policy loss over $\gamma$ (lower = better). Shown: mean and standard errors over 5 runs.

**Experiments with estimated nuisance functions.** We now consider two settings to analyze the effectiveness of our algorithm when using estimated nuisance functions in stage 1. In setting A, we consider a synthetic dataset with nonlinear CATE and aim to learn a linear $g$ (similar to Fig. 1). In setting B, we consider a non-linear but regularized $g$ (similar to Fig. 2). Details regarding the datasets are in Appendix D.

**Results.** We report PEHE and policy loss for all four pseudo-outcomes (PI, RA, IPW, and DR) in Fig. 4 (Setting A) and Fig. 5 (Setting B). The results demonstrate that our algorithm is effective in decreasing the policy loss compared to the baselines ($\gamma = 0$) when increasing $\gamma$. The results for IPW and DR are more noisy as compared to the ones for PI and RA, which is likely due to higher variance as a result of divisions by propensity scores (a known issue for these estimation methods; see [9]). Nevertheless, our PT-CATE algorithm leads to a better average decision performance across all estimation methods while only minimally increasing the PEHE.

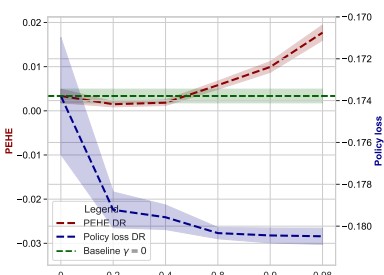

Figure 6: **Experimental results for real-world data.** Shown: PEHE and policy loss over $\gamma$ (lower = better). Shown: Mean and $80\%$ confidence intervals over 5 runs.

*Real-world data.* **Dataset.** Here, we provide additional experimental results using the *Hillstrom Email Marketing dataset* of $n = 64000$ customers. Details regarding the dataset and our preprocessing are in Appendix E.

**Results.** Similar to the experiments with simulated data in Fig. 4 and Fig. 5, we plot the (estimated) PEHE and policy loss over different values of $\gamma$ and compare it to the baseline CATE $\gamma = 0$ (here: for the DR-learner). The results are shown in Fig 6. The results are consistent with our experiments on synthetic data: our algorithm is effective in improving the policy loss as compared to standard CATE estimation ($\gamma = 0$).

Table 1: Improvement of our PT-CATE-based policy over the observational policy.

| $\gamma$ | Policy value | Improv. | Improv. (%) |
|---|---|---|---|
| Obs. policy | $1.450 \pm 0.000$ | — | — |
| $\gamma = 0$ | $1.738 \pm 0.045$ | 0.029 | 19.827 |
| $\gamma = 0.2$ | $1.792 \pm 0.014$ | 0.034 | 23.579 |
| $\gamma = 0.4$ | $1.796 \pm 0.009$ | 0.035 | 23.823 |
| $\gamma = 0.8$ | $1.803 \pm 0.004$ | 0.035 | 24.346 |
| $\gamma = 0.9$ | $1.804 \pm 0.006$ | 0.035 | 24.423 |
| $\gamma = 0.98$ | $1.805 \pm 0.006$ | 0.035 | 24.451 |

Reported: policy values (mean $\pm$ std dev)$\times 10$ and average improvement over 5 seeds.

We also compare against the behavioral policy that generated the data (i.e., using the propensity score $\pi_b$ as a policy). For this, we report the improvement over the behavioral policy for different values of $\gamma$ in Table 1. As we can see, using our PT-CATE algorithm with $\gamma = 0.98$ can lead to a 24.45% improved response probability as compared to just using a CATE-based policy ($\gamma = 0$).

## 6 Discussion

In this paper, we showed that standard two-stage CATE estimators can be suboptimal for decision-making and propose a policy-targeted CATE (PT-CATE) to balance estimation and decision performance. Our neural algorithm improves CATE for decision-making while maintaining interpretability as CATE.

**Limitations:** If the second-stage model class is not restricted, our method will not lead to improvement over existing two-stage learners. However, it will also not introduce additional bias as the PT-CATE simplifies to standard CATE.

**Societal risks:** As with any causal inference methods, there are risks of misuse if applied without proper understanding of underlying assumptions or in contexts with significant unmeasured confounding. Additionally, automated decision systems based on our approach could perpetuate or amplify existing biases if training data reflects historical inequities.

**Future work:** Future directions may include extensions to other settings, such as time series and reinforcement learning (e.g., Q-learning), as well as real-world validation in healthcare and public policy. Furthermore, one may consider incorporating uncertainty into our method using e.g., Bayesian approaches. Finally, methods for variance reduction such as stabilized weighting or propensity clipping may be employed to improve performance in practice.

**Conclusion:** In sum, our method provides practitioners with a principled tool for reliable, data-driven decision-making by improving the decision performance of two-stage meta-learners under model-class restrictions.

## Acknowledgements

This paper is supported by the DAAD program "Konrad Zuse Schools of Excellence in Artificial Intelligence", sponsored by the Federal Ministry of Education and Research. Dennis Frauen gratefully acknowledges financial support from G-Research.

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

# A  Extended related work

## A.1  Deep learning for CATE estimation

In recent years, deep neural networks have gained considerable traction for estimating the conditional average treatment effect (CATE) due to scalability reasons and the ability to extract complex features from multimodal data. Notable advances include methods for learning representations that improve CATE estimation through balancing techniques [23, 50, 62], disentanglement strategies [20], and the incorporation of inductive biases [9, 10].

While these approaches focus on estimating the nuisance functions $\eta$, they do not estimate CATE directly. Instead, they are compatible with two-stage meta-learners, including the algorithm proposed in this work. Importantly, existing approaches prioritize accuracy in CATE estimation rather than exploring the interplay between CATE estimation and decision-making, which remains a gap in the existing literature.

## A.2  Orthogonal learning

Orthogonal learning (also called debiased learning) is rooted in semiparametric efficiency theory, which has become widespread in estimating heterogeneous treatment effects [57, 56]. These learners are designed to be robust against errors in nuisance function estimation and offer strong theoretical guarantees.

For CATE estimation, orthogonal learning has been proposed by [8, 16]. Specific instantiations are, for instance, the DR-learner [55, 33] and the R-learner [46], where the latter can be interpreted within the broader context of overlap-weighted DR estimators [42]. Orthogonal learners are, by construction, two-stage meta-learners and thus are applicable to our proposed framework. In this work, we leverage the DR-learner outlined in Eq. (10), but extensions to the R-learner/ overlap-weighted methods could be of interest for future research.

## A.3  Dynamic settings

Both off-policy learning (OPL) and CATE estimation have a well-established history in dynamic settings, where treatments are administered, and outcomes are observed over time. In the context of CATE estimation, methods have been developed for both model-based approaches [40, 6, 41] and model-agnostic techniques [17]. For OPL, dynamic treatment regimes have been extensively studied [43, 44, 60, 28], alongside reinforcement learning approaches for Markov decision processes under stationarity assumptions [22, 53, 29, 26, 27]. Extending our approach to dynamic settings presents an intriguing avenue for future work. Such extensions could address challenges unique to temporal data, including time-varying confounding.

# B Proofs

## B.1 Proof of Theorem 4.1

*Proof.* Let $\mathcal{S}$ be the set of step functions

$$\mathcal{S} = \left\{ f : [a,b] \to \mathbb{R} \;\middle|\; \begin{array}{l} \exists\, n \in \mathbb{N},\; \exists\, a = x_0 < x_1 < \cdots < x_n = b, \\ \exists\, c_1, c_2, \ldots, c_n \in \mathbb{R} \text{ such that } f(x) = \sum_{i=1}^{n} c_i\, \mathbf{1}(x \in [x_{i-1}, x_i)) \end{array} \right\}. \tag{14}$$

Let $\mathcal{S}_{\mathcal{G}}$ denote the set of step functions $\tau$ such that the optimal approximation $g^\tau$ gets the sign correct everywhere, i.e.,

$$\mathcal{S}_{\mathcal{G}} = \left\{ \tau \in \mathcal{S} \;\middle|\; \mathrm{sgn}(g_\tau(x)) = \mathrm{sgn}(\tau(x)) \text{ for all } x \text{ and } g_\tau \in \arg\min_{g \in \mathcal{G}} \mathbb{E}[(\tau(X) - g(X))^2] \right\}. \tag{15}$$

Note that $\mathcal{S}_{\mathcal{G}}$ is non-empty as any constant positive function is in $\mathcal{S}_{\mathcal{G}}$.

For any $\tau \in \mathcal{S}$, we define the supremum norm as $\|\tau\|_\infty = \sup_{x \in [a,b]} |\tau(x)|$. We set

$$M := \sup_{\tau \in \mathcal{S}_{\mathcal{G}}} \|\tau\|_\infty. \tag{16}$$

By definition of the supremum, there exists a sequence $\{\tau_n\}_{n \geq 1} \subset \mathcal{S}_{\mathcal{G}}$ such that

$$\|\tau_n\|_\infty \to M \quad \text{as } n \to \infty. \tag{17}$$

Hence, for any fixed $\epsilon > 0$, one can select an index $n_\epsilon$ such that

$$\|\tau_{n_\epsilon}\|_\infty > M - \epsilon. \tag{18}$$

Therefore, there exists a corresponding approximation

$$g_{\tau_{n_\epsilon}} \in \arg\min_{g \in \mathcal{G}} \mathbb{E}\big[(\tau_{n_\epsilon}(X) - g(X))^2\big], \tag{19}$$

satisfying

$$\mathrm{sgn}\big(g_{\tau_{n_\epsilon}}(x)\big) = \mathrm{sgn}\big(\tau_{n_\epsilon}\big) \quad \text{for all } x \in [a,b], \tag{20}$$

which implies that the corresponding thresholded policy $\pi_{g_{\tau_{n_\epsilon}}}$ is optimal, i.e.,

$$\pi_{g_{\tau_{n_\epsilon}}} \in \arg\max_{\pi \in \Pi_{\mathcal{G}}} V_{\tau_{n_\epsilon}}(\pi). \tag{21}$$

Because $\tau_{n_\epsilon} \in \mathcal{S}_{\mathcal{G}}$ is a step function, there exists an interval $I_{\tau_{n_\epsilon}} \subseteq [a,b]$ of positive measure so that

$$|\tau_{n_\epsilon}(x)| = \|\tau_{n_\epsilon}^*\|_\infty > M - \epsilon \quad \text{for all } x \in I_{\tau_{n_\epsilon}} \tag{22}$$

on which $|\tau^*(x)|$ is nearly $\|\tau^*\|_\infty$.

Let $\epsilon > 0$ be fixed, and let $\delta > \epsilon$. We define the CATE $\tau^*$ as

$$\tau^*(x) = \begin{cases} \tau_{n_\epsilon}(x) + \delta\, \mathrm{sgn}\big(\tau_{n_\epsilon}(x)\big), & \text{if } x \in I, \\ \tau_{n_\epsilon}(x), & \text{if } x \notin I. \end{cases} \tag{23}$$

Then, by definition of $\tau^*$, we have

$$\|\tau^*\|_\infty > M, \tag{24}$$

which implies by definition of $M$ that $\tau^*$ satisfies

$$\tau^* \notin \mathcal{S}_{\mathcal{G}}. \tag{25}$$

That is, the optimal approximation of $g_{\tau^*}$ with functions in $\mathcal{G}$ must fail to preserve the sign of $\tau^*$ on a subset of positive measure $\mathcal{E}$, i.e.,

$$\mathrm{sgn}\big(g_{\tau^*}(x)\big) \neq \mathrm{sgn}\big(\tau^*(x)\big) \quad \text{for all } x \in \mathcal{E}. \tag{26}$$

We now compare the plug-in policy induced by $g_{\tau^*}$, i.e.,

$$\pi_{g_{\tau^*}}(x) = \mathbf{1}\left(g_{\tau^*}(x) > 0\right), \tag{27}$$

against the policy obtained by thresholding an approximation of $\tau_{n_\epsilon}(x)$, i.e.,

$$\pi_{g_{\tau_{n_\epsilon}}}(x) = \mathbf{1}\left(g_{\tau_{n_\epsilon}}(x) > 0\right). \tag{28}$$

On the set $\mathcal{E}$, the decisions made by $\pi_{g_{\tau_\epsilon}}$ and $\pi^*_{\tau_{n_\epsilon}}$ differ in a way such that

$$V_{\tau^*}\left(\pi_{g_{\tau^*}}\right) < V_{\tau^*}\left(\pi_{g_{\tau_{n_\epsilon}}}\right). \tag{29}$$

Furthermore, $\tau_{n_\epsilon}$ and $\tau^*$ have the same sign by definition, which implies together with Eq. (21) that

$$\pi_{g_{\tau_{n_\epsilon}}} \in \arg\max_{\pi \in \Pi_\mathcal{G}} V_{\tau^*}(\pi), \tag{30}$$

because $\tau_{n_\epsilon} \in \mathcal{S}_\mathcal{G}$ and thus $g_{\tau_{n_\epsilon}} \in \Pi_\mathcal{G}$. Hence,

$$V_{\tau^*}\left(\pi_{g_{\tau^*}}\right) < \max_{\pi \in \Pi_\mathcal{G}} V_{\tau^*}(\pi), \tag{31}$$

which completes the proof. $\qquad\square$

## B.2 Proof of Theorem 4.3

*Proof.* One can show that, for all pseudo-outcomes $Y_\eta^m$, it holds that

$$\mathbb{E}\left[Y_\eta^m \mid X\right] = \tau(X) \tag{32}$$

(see e.g., [9] for a proof). Hence, we can apply the tower property and write

$$\mathcal{L}_{\gamma,\alpha,\eta}^m(g) = (1-\gamma)\,\mathbb{E}\left[\left(Y_\eta^m - g(X)\right)^2\right] - \gamma\,\mathbb{E}\left[Y_\eta^m \sigma\left(\alpha(X)g(X)\right)\right] \tag{33}$$

$$= (1-\gamma)\,\mathbb{E}\left[\left(Y_\eta^m - \tau(X) + \tau(X) - g(X)\right)^2\right] - \gamma\,\mathbb{E}\left[\mathbb{E}\left[Y_\eta^m \sigma\left(\alpha(X)g(X)\right)\big|X\right]\right] \tag{34}$$

$$\propto (1-\gamma)\,\mathbb{E}\left[\left(\tau(X) - g(X)\right)^2 + 2\left(\tau(X) - g(X)\right)\left(Y_\eta^m - \tau(X)\right)\right] \tag{35}$$
$$\quad - \gamma\,\mathbb{E}\left[\mathbb{E}\left[\tau(X)\sigma\left(\alpha(X)g(X)\right)\big|X\right]\right]$$

$$= (1-\gamma)\,\mathbb{E}\left[\mathbb{E}\left[\left(\tau(X) - g(X)\right)^2 + 2\left(\tau(X) - g(X)\right)\left(Y_\eta^m - \tau(X)\right)\right]\big|X\right] \tag{36}$$
$$\quad - \gamma\,\mathbb{E}\left[\tau(X)\sigma\left(\alpha(X)g(X)\right)\right]$$

$$= (1-\gamma)\,\mathbb{E}\left[\left(\tau(X) - g(X)\right)^2\right] - \gamma\,\mathbb{E}\left[\tau(X)\sigma\left(\alpha(X)g(X)\right)\right] \tag{37}$$

$$= \mathcal{L}_{\gamma,\alpha}^m(g). \tag{38}$$

$$\qquad\square$$

## B.3 Theoretical result with nuisance errors (Theorem 4.4)

In the following, we provide a new, slightly stronger theoretical result than in Theorem 4.4 but which guarantees that minimizing our proposed loss results in a reasonable PT-CATE estimator, even when the nuisance functions are estimated with errors. Importantly, we upper bound of the PT-CATE error on the nuisance errors of the respective adjustment method (pseudo-outcome). For the DR pseudo-outcome, **we establish a doubly robust convergence rate**.

**Theorem B.1.** *Let $g^* = \arg\min_{g \in \mathcal{G}} \mathcal{L}_{\gamma,\alpha,\eta}^m(g)$ and $\hat{g} = \arg\min_{g \in \mathcal{G}} \mathcal{L}_{\gamma,\alpha,\hat{\eta}}^m(g)$ be the minimizers of the PT-CATE loss with ground-truth and estimated nuisances for a fixed indicator approximation $\alpha$ and $\gamma \in [0,1]$. We assume the following regularity condition: there exists a constant $\delta > 0$, so that, for all $\bar{g} \in \mathrm{star}(\mathcal{G}, g^*) = \{tg^* + (1-t)g | g \in \mathcal{G}\}$, it holds that*

$$\frac{\mathbb{E}\left[-Y_{\hat{\eta}}^m \sigma''\left(\alpha(X)\bar{g}(X)\right)\alpha(X)^2(\hat{g}(X) - g^*(X))^2\right]}{||g^* - \hat{g}||^2} \geq \delta, \tag{39}$$

*where $||g^* - \hat{g}||^2 = \mathbb{E}[(g^*(X) - \hat{g}(X))^2]$ denotes the squared $L_2$-norm and $\sigma''(\cdot)$ denotes the second derivative of the sigmoid function. Furthermore, assume that the propensity estimator and*

*ground-truth response functions are bounded via $p \leq \hat{\pi}(x) \leq 1 - p$ and $|\mu_a(x)| \leq c$ for constants $p, c > 0$ and for all $x \in \mathcal{X}$.*

*Then, for all $\rho_1, \rho_2 > 0$ so that $(1 - \gamma)\rho_1 + \frac{\gamma}{2}\rho_2 < 1 - \gamma + \frac{\delta}{2}\gamma$, it holds that*

$$||g^* - \hat{g}||^2 \leq \frac{R_{\gamma,\alpha,\hat{\eta}}^m(\hat{g}, g^*) + M_{\hat{\eta},\eta}^m \left( \frac{(1-\gamma)}{\rho_1} + \gamma \frac{C_\alpha}{2\rho_2} \right)}{1 - \rho_1 + \gamma \left( \frac{\delta}{2} - 1 + \rho_1 - \frac{\rho_2}{2} \right)}, \tag{40}$$

*where $R_{\gamma,\alpha,\hat{\eta}}^m(\hat{g}, g^*) = \mathcal{L}_{\gamma,\alpha,\hat{\eta}}^m(\hat{g}) - \mathcal{L}_{\gamma,\alpha,\hat{\eta}}^m(g^*)$ is an optimization-dependent term, $C_\alpha > 0$ is a constant depending on $\alpha$, and $M_{\hat{\eta},\eta}^m$ is the (pseudo-outcome-dependent) rate term, defined via*

$$M_{\hat{\eta},\eta}^{\mathrm{PI}} = M_{\hat{\eta},\eta}^{\mathrm{RA}} = 2||\hat{\mu}_1 - \mu_1||^2 + 2||\hat{\mu}_0 - \mu_0||^2 \tag{41}$$

$$M_{\hat{\eta},\eta}^{\mathrm{IPW}} = \frac{c^2}{p^2}||\hat{\pi}_b - \pi_b||^2 \tag{42}$$

$$M_{\hat{\eta},\eta}^{\mathrm{DR}} = \frac{2}{p^2}||\hat{\pi}_b - \pi_b||^2 \left( ||\hat{\mu}_1 - \mu_1||^2 + ||\hat{\mu}_0 - \mu_0||^2 \right). \tag{43}$$

*Proof.* Recall that

$$\mathcal{L}_{\gamma,\alpha,\eta}^m(g) = (1 - \gamma)\mathbb{E}[(Y_\eta^m - g(X))^2] - \gamma \mathbb{E}\left[ Y_\eta^m \sigma\left( \alpha(X)g(X) \right) \right] \tag{44}$$

$$= (1 - \gamma)\mathcal{L}_{\eta,\mathrm{MSE}}^m(g) - \gamma \mathcal{L}_{\eta,\alpha}^m(g). \tag{45}$$

We can write

$$\mathcal{L}_{\hat{\eta},\mathrm{MSE}}^m(\hat{g}) = \mathbb{E}[(Y_{\hat{\eta}}^m - g^*(X) + g^*(X) - \hat{g}(X))^2] \tag{46}$$

$$= \mathcal{L}_{\hat{\eta},\mathrm{MSE}}^m(g^*) + ||g^* - \hat{g}||^2 - 2\mathbb{E}\left[ (Y_{\hat{\eta}}^m - g^*(X))(\hat{g}(X) - g^*(X)) \right]. \tag{47}$$

For $\mathcal{L}_{\hat{\eta},\alpha}^m(\hat{g})$, we can do a functional Taylor expansion, i.e., there exists a $\bar{g} \in \mathrm{star}(\mathcal{G}, g^*)$ with

$$\mathcal{L}_{\hat{\eta},\alpha}^m(\hat{g}) = \mathcal{L}_{\hat{\eta},\alpha}^m(g^*) + D_g\mathcal{L}_{\hat{\eta},\alpha}^m(g^*)[\hat{g} - g^*] + \frac{1}{2}D_gD_g\mathcal{L}_{\hat{\eta},\alpha}^m(\bar{g})[\hat{g} - g^*, \hat{g} - g^*], \tag{48}$$

where $D_g$ denotes the functional derivative.

For the first-order derivative, we obtain

$$D_g\mathcal{L}_{\hat{\eta},\alpha}^m(g^*)[\hat{g} - g^*] = \frac{d}{dt}\mathbb{E}\left[ Y_{\hat{\eta}}^m \sigma\left( \alpha(X)(g^*(X) + t(\hat{g}(X) - g^*(X))) \right) \right]\Big|_{t=0} \tag{49}$$

$$= \mathbb{E}\left[ Y_{\hat{\eta}}^m \sigma'\left( \alpha(X)g^*(X) \right) \alpha(X)(\hat{g}(X) - g^*(X)) \right], \tag{50}$$

where $\sigma'(\cdot)$ denotes the derivative of the sigmoid function.

For the second-order derivative, we obtain

$$D_gD_g\mathcal{L}_{\hat{\eta},\alpha}^m(\bar{g})[\hat{g} - g^*, \hat{g} - g^*] \tag{51}$$

$$= \frac{d^2}{dt d\nu}\mathbb{E}\left[ Y_{\hat{\eta}}^m \sigma\left( \alpha(X)(\bar{g}(X) + t(\hat{g}(X) - g^*(X)) + \nu(\hat{g}(X) - g^*(X))) \right) \right]\Big|_{t=\nu=0} \tag{52}$$

$$= \frac{d}{dt}\mathbb{E}\left[ Y_{\hat{\eta}}^m \sigma'\left( \alpha(X)(\bar{g}(X) + t(\hat{g}(X) - g^*(X))) \right) \alpha(X)(\hat{g}(X) - g^*(X)) \right]\Big|_{t=0} \tag{53}$$

$$= \mathbb{E}\left[ Y_{\hat{\eta}}^m \sigma''\left( \alpha(X)\bar{g}(X) \right) \alpha(X)^2(\hat{g}(X) - g^*(X))^2 \right] \tag{54}$$

$$\leq -\delta||g^* - \hat{g}||^2, \tag{55}$$

where the last inequality follows from the regularity assumption.

Putting everything together, we obtain that

$$\mathcal{L}_{\gamma,\alpha,\hat{\eta}}^m(\hat{g}) \geq (1 - \gamma)\left( \mathcal{L}_{\hat{\eta},\mathrm{MSE}}^m(g^*) + ||g^* - \hat{g}||^2 - 2\mathbb{E}\left[ (Y_{\hat{\eta}}^m - g^*(X))(\hat{g}(X) - g^*(X)) \right] \right) \tag{56}$$

$$- \gamma \left( \mathcal{L}_{\hat{\eta},\alpha}^m(g^*) + \mathbb{E}\left[ Y_{\hat{\eta}}^m \sigma'\left( \alpha(X)g^*(X) \right) \alpha(X)(\hat{g}(X) - g^*(X)) \right] - \frac{\delta}{2}||g^* - \hat{g}||^2 \right) \tag{57}$$

or equivalently

$$(1 - \gamma + \frac{\delta}{2}\gamma)||g^* - \hat{g}||^2 \leq R_{\gamma,\alpha,\hat{\eta}}^m(\hat{g}, g^*) + 2(1 - \gamma)\mathbb{E}\left[(Y_{\hat{\eta}}^m - g^*(X))(\hat{g}(X) - g^*(X))\right] \quad (58)$$

$$+ \gamma\mathbb{E}\left[Y_{\hat{\eta}}^m \sigma'\left(\alpha(X)g^*(X)\right)\alpha(X)(\hat{g}(X) - g^*(X))\right], \quad (59)$$

where $R_{\gamma,\alpha,\hat{\eta}}^m(\hat{g}, g^*) = \mathcal{L}_{\gamma,\alpha,\hat{\eta}}^m(\hat{g}) - \mathcal{L}_{\gamma,\alpha,\hat{\eta}}^m(g^*)$.

$$\mathbb{E}\left[(Y_{\hat{\eta}}^m - g^*(X))(\hat{g}(X) - g^*(X))\right] = \mathbb{E}\left[(Y_{\hat{\eta}}^m - Y_{\eta}^m)(\hat{g}(X) - g^*(X))\right] \quad (60)$$

$$+ \mathbb{E}\left[(Y_{\eta}^m - g^*(X))(\hat{g}(X) - g^*(X))\right] \quad (61)$$

$$= \mathbb{E}\left[(\Delta^m(X)(\hat{g}(X) - g^*(X))\right] \quad (62)$$

$$+ \mathbb{E}\left[(\tau(X) - g^*(X))(\hat{g}(X) - g^*(X))\right] \quad (63)$$

$$\quad (64)$$

with $\Delta^m(X) = \mathbb{E}[Y_{\hat{\eta}}^m - Y_{\eta}^m | X]$. Similarly,

$$\mathbb{E}\left[Y_{\hat{\eta}}^m \sigma'\left(\alpha(X)g^*(X)\right)\alpha(X)(\hat{g}(X) - g^*(X))\right] \quad (65)$$

$$= \mathbb{E}\left[\Delta^m(X)\sigma'\left(\alpha(X)g^*(X)\right)\alpha(X)(\hat{g}(X) - g^*(X))\right] \quad (66)$$

$$+ \mathbb{E}\left[\tau(X)\sigma'\left(\alpha(X)g^*(X)\right)\alpha(X)(\hat{g}(X) - g^*(X))\right]. \quad (67)$$

Putting everything together, we obtain

$$(1 - \gamma + \frac{\delta}{2}\gamma)||g^* - \hat{g}||^2 \leq R_{\gamma,\alpha,\hat{\eta}}^m(\hat{g}, g^*) + 2(1 - \gamma)\mathbb{E}\left[(\Delta^m(X)(\hat{g}(X) - g^*(X))\right] \quad (68)$$

$$+ 2(1 - \gamma)\mathbb{E}\left[(\tau(X) - g^*(X))(\hat{g}(X) - g^*(X))\right] \quad (69)$$

$$+ \gamma\mathbb{E}\left[\Delta^m(X)\sigma'\left(\alpha(X)g^*(X)\right)\alpha(X)(\hat{g}(X) - g^*(X))\right] \quad (70)$$

$$+ \gamma\mathbb{E}\left[\tau(X)\sigma'\left(\alpha(X)g^*(X)\right)\alpha(X)(\hat{g}(X) - g^*(X))\right]. \quad (71)$$

Note that

$$2(1 - \gamma)\mathbb{E}\left[(\tau(X) - g^*(X))(\hat{g}(X) - g^*(X))\right] \quad (72)$$

$$+ \gamma\mathbb{E}\left[\tau(X)\sigma'\left(\alpha(X)g^*(X)\right)\alpha(X)(\hat{g}(X) - g^*(X))\right] \quad (73)$$

$$= -(1 - \gamma)D_g\mathcal{L}_{\eta,\text{MSE}}^m(g^*)[\hat{g} - g^*] - \gamma D_g\mathcal{L}_{\eta,\alpha}^m(g^*)[\hat{g} - g^*] \quad (74)$$

$$= -D_g\mathcal{L}_{\gamma,\alpha,\eta}^m(g^*)[\hat{g} - g^*] \quad (75)$$

$$\leq 0 \quad (76)$$

because $g^*$ is a minimizer of the oracle nuisance loss $\mathcal{L}_{\gamma,\alpha,\eta}^m$. Hence,

$$(1 - \gamma + \frac{\delta}{2}\gamma)||g^* - \hat{g}||^2 \leq R_{\gamma,\alpha,\hat{\eta}}^m(\hat{g}, g^*) + 2(1 - \gamma)\mathbb{E}\left[(\Delta^m(X)(\hat{g}(X) - g^*(X))\right] \quad (77)$$

$$+ \gamma\mathbb{E}\left[\Delta^m(X)\sigma'\left(\alpha(X)g^*(X)\right)\alpha(X)(\hat{g}(X) - g^*(X))\right]. \quad (78)$$

For the different pseudo outcomes, we can write

$$\Delta^{\text{PI}}(X) = \hat{\mu}_1(X) - \mu_1(X) + \hat{\mu}_0(X) - \mu_0(X) \quad (79)$$

$$\Delta^{\text{RA}}(X) = \pi_b(X)(\mu_0(X) - \hat{\mu}_0(X)) + (1 - \pi_b(X))(\hat{\mu}_1(X) - \mu_1(X)) \quad (80)$$

$$\Delta^{\text{IPW}}(X) = \frac{\mu_1(X)}{\hat{\pi}_b(X)}(\pi_b(X) - \hat{\pi}_b(X)) - \frac{\mu_0(X)}{1 - \hat{\pi}_b(X)}(\hat{\pi}_b(X) - \pi_b(X)) \quad (81)$$

$$\Delta^{\text{DR}}(X) = \frac{1}{\hat{\pi}_b(X)}(\pi_b(X) - \hat{\pi}_b(X))(\hat{\mu}_1(X) - \mu_1(X)) - \frac{1}{1 - \hat{\pi}_b(X)}(\hat{\pi}_b(X) - \pi_b(X))(\mu_0(X) - \hat{\mu}_0(X))$$

$$\quad (82)$$

By applying the Cauchy-Schwarz inequality, we obtain

$$\mathbb{E}\left[(\Delta^{\mathrm{PI}}(X)(\hat{g}(X) - g^*(X))\right] \leq ||\hat{g} - g^*|| \left(||\hat{\mu}_1 - \mu_1|| + ||\hat{\mu}_0 - \mu_0||\right) \tag{83}$$

$$\mathbb{E}\left[(\Delta^{\mathrm{RA}}(X)(\hat{g}(X) - g^*(X))\right] \leq ||\hat{g} - g^*|| \left(||\hat{\mu}_1 - \mu_1|| + ||\hat{\mu}_0 - \mu_0||\right) \tag{84}$$

$$\mathbb{E}\left[(\Delta^{\mathrm{IPW}}(X)(\hat{g}(X) - g^*(X))\right] \leq ||\hat{g} - g^*|| \frac{c}{p} ||\hat{\pi}_b - \pi_b|| \tag{85}$$

$$\mathbb{E}\left[(\Delta^{\mathrm{DR}}(X)(\hat{g}(X) - g^*(X))\right] \leq ||\hat{g} - g^*|| \left(\frac{1}{p}||\hat{\pi}_b - \pi_b|| \left(||\hat{\mu}_1 - \mu_1|| + ||\hat{\mu}_0 - \mu_0||\right)\right). \tag{86}$$

By applying AM-GM inequality and the fact that $(a + b)^2 \leq 2(a^2 + b^2)$, it holds for any $\rho > 0$ that

$$2\mathbb{E}\left[(\Delta^{\mathrm{PI}}(X)(\hat{g}(X) - g^*(X))\right] \leq \rho_1 ||\hat{g} - g^*||^2 + \frac{2}{\rho_1}\left(||\hat{\mu}_1 - \mu_1||^2 + ||\hat{\mu}_0 - \mu_0||^2\right) \tag{87}$$

$$2\mathbb{E}\left[(\Delta^{\mathrm{RA}}(X)(\hat{g}(X) - g^*(X))\right] \leq \rho_1 ||\hat{g} - g^*||^2 + \frac{2}{\rho_1}\left(||\hat{\mu}_1 - \mu_1||^2 + ||\hat{\mu}_0 - \mu_0||^2\right) \tag{88}$$

$$2\mathbb{E}\left[(\Delta^{\mathrm{IPW}}(X)(\hat{g}(X) - g^*(X))\right] \leq \rho_1 ||\hat{g} - g^*||^2 + \frac{c^2}{\rho_1 p^2} ||\hat{\pi}_b - \pi_b||^2 \tag{89}$$

$$2\mathbb{E}\left[(\Delta^{\mathrm{DR}}(X)(\hat{g}(X) - g^*(X))\right] \leq \rho_1 ||\hat{g} - g^*||^2 + \frac{2}{\rho_1 p^2} ||\hat{\pi}_b - \pi_b||^2 \left(||\hat{\mu}_1 - \mu_1||^2 + ||\hat{\mu}_0 - \mu_0||^2\right). \tag{90}$$

We can write this in generalized form via

$$2\mathbb{E}\left[(\Delta^{\mathrm{m}}(X)(\hat{g}(X) - g^*(X))\right] \leq \rho_1 ||\hat{g} - g^*||^2 + \frac{1}{\rho_1} M^m_{\hat{\eta}, \eta}. \tag{91}$$

Using the same arguments and the fact that we can upperbound $\sigma'\left(\alpha(X)g^*(X)\right)^2 \alpha(X)^2 \leq C_\alpha$ for some constant $C_\alpha > 0$, we obtain

$$\mathbb{E}\left[\Delta^m(X)\sigma'\left(\alpha(X)g^*(X)\right)\alpha(X)(\hat{g}(X) - g^*(X))\right] \leq \frac{\rho_2}{2}||\hat{g} - g^*||^2 + \frac{C_\alpha}{2\rho_2} M^m_{\hat{\eta}, \eta}. \tag{92}$$

Hence, it holds that

$$(1 - \gamma + \frac{\delta}{2}\gamma)||g^* - \hat{g}||^2 \leq R^m_{\gamma, \alpha, \hat{\eta}}(\hat{g}, g^*) + (1 - \gamma)\left(\rho_1 ||\hat{g} - g^*||^2 + \frac{1}{\rho_1} M^m_{\hat{\eta}, \eta}\right) \tag{93}$$

$$+ \gamma\left(\frac{\rho_2}{2}||\hat{g} - g^*||^2 + \frac{C_\alpha}{2\rho_2} M^m_{\hat{\eta}, \eta}\right), \tag{94}$$

or, equivalently, for $\rho_1, \rho_2$ so that $(1 - \gamma)\rho_1 + \frac{\gamma}{2}\rho_2 < 1 - \gamma + \frac{\delta}{2}\gamma$, we have

$$||g^* - \hat{g}||^2 \leq \frac{R^m_{\gamma, \alpha, \hat{\eta}}(\hat{g}, g^*) + (1 - \gamma)\frac{1}{\rho_1} M^m_{\hat{\eta}, \eta} + \gamma\frac{C_\alpha}{2\rho_2} M^m_{\hat{\eta}, \eta}}{1 - \gamma + \frac{\delta}{2}\gamma - (1 - \gamma)\rho_1 - \frac{\gamma}{2}\rho_2} \tag{95}$$

$$= \frac{R^m_{\gamma, \alpha, \hat{\eta}}(\hat{g}, g^*) + M^m_{\hat{\eta}, \eta}\left(\frac{(1 - \gamma)}{\rho_1} + \gamma\frac{C_\alpha}{2\rho_2}\right)}{1 - \rho_1 + \gamma\left(\frac{\delta}{2} - 1 + \rho_1 - \frac{\rho_2}{2}\right)}. \tag{96}$$

$\square$

## B.4 Additional result bounding policy value discrency with projection error

If the projection error is small (i.e., our learned CATE is close to the ground-truth CATE), we are able to bound the discrepancy in policy value as follows.

**Lemma B.2.** *Let the true CATE be $\tau(x) = \mu_1(x) - \mu_0(x) \in L^2(\mathbb{P})$. For a function class $\mathcal{G} \subset L^2(\mathbb{P})$, define its $L^2$-projection of $\tau$ as*

$$g^\star = \arg\min_{g \in \mathcal{G}} \mathbb{E}\left[(\tau(X) - g(X))^2\right],$$

*and let $\varepsilon_2 = \|\tau - g^\star\|_2$. Furthermore, introduce the corresponding thresholded treatment policies*

$$\pi_\tau(x) = \mathbb{1}\{\tau(x) > 0\}, \qquad \pi_{g^\star}(x) = \mathbb{1}\{g^\star(x) > 0\},$$

*and the (shifted) policy value*

$$V(\pi) = \mathbb{E}[\pi(X)\,\tau(X)].$$

*Then it holds that*

$$0 \;\leq\; V(\pi_\tau) - V(\pi_{g^\star}) \;\leq\; \varepsilon_2.$$

*Proof.* Define the disagreement set

$$\mathcal{M} \;:=\; \{x : \operatorname{sign}(\tau(x)) \neq \operatorname{sign}(g^\star(x))\}.$$

Then

$$\Delta := V(\pi_\tau) - V(\pi_{g^\star}) = \mathbb{E}\big[\tau(X)\big(\pi_\tau(X) - \pi_{g^\star}(X)\big)\big] = \mathbb{E}[\tau(X)\,\mathbf{1}_{\mathcal{M}}(X)].$$

On $\mathcal{M}$ we have $\tau(X)\,g^\star(X) \leq 0$, hence $|\tau(X)| \leq |\tau(X) - g^\star(X)|$. Therefore

$$\Delta \leq \mathbb{E}[\,|\tau(X) - g^\star(X)|\,\mathbf{1}_{\mathcal{M}}(X)] \leq \mathbb{E}[\,|\tau(X) - g^\star(X)|\,] = \|\tau - g^\star\|_1 \leq \varepsilon_2,$$

where the last inequality follows from Hölder's inequality. $\qquad\square$

## C  Implementation details

**Estimation of nuisance functions** We estimate all nuisance functions $\mu_1$, $\mu_0$, and $\pi_b$ with standard feed-forward neural networks using 4 layers with tanh activations. The response functions $\mu_a$ are regression functions, which we fit by minimizing the MSE loss on the filtered datasets where we condition on $A = a$. Estimating the propensity score $\pi_b$ is a classification task so that we apply a sigmoid output activation function and minimize the binary cross entropy loss. For the synthetic experiments, we mimic randomized controlled trials (RCTs) and use the ground-truth propensity score which we assume to be known.

**Second-stage model:** For our second-stage model (Fig. 3), we model each of $g_\theta$ and $\alpha_\phi$ as feed-forward neural networks with 4 layers with tanh activations for $g_\theta$ and ReLU activations for $\alpha_\phi$. For the experiments with linear $g_\theta$ (i.e., Fig. 2 and Fig. 4), we set $g_\theta$ to a single linear layer. For the experiments with regularized $g_\theta$ (i.e., Fig. 1 and Fig. 5), we choose a custom regularization parameter for each pseudo-outcome type that yields a misspecified initial CATE estimate. For the synthetic experiments in Fig 4 and Fig 5, we normalize $g_\theta$ by applying a tanh output activation in step 2 of our learning algorithm (Algorithm 1) as we observed that this can stabilize the optimization of $\alpha_\phi$. We also applied a weighting scheme in step 3 via $1/\alpha_\phi(X)$ to further encourage sharp indicator approximation of regions where the initial CATE is correct. We ran our algorithm for $K = 1$ iteration.

**Hyperparameters.** To ensure a fair comparison, we use the same hyperparameters for each second-stage learner across different $\gamma$ and random seeds. For reproducibility purposes, we report the hyperparameters used (e.g., dimensions, learning rate) for all experiments and models (including nuisance functions) as `.yaml` files.[3]

**Runtime.** For the second-stage models, training took approximately two minutes using $n = 2000$ samples and a standard computer with AMD Ryzen 7 Pro CPU and 32GB of RAM.

**Full learning algorithm.** The full learning algorithm is reported in Algorithm 1 below.

---

**Algorithm 1:** Re-targeted CATE estimation (PT-CATE)

---

1: **Input:** Training data $\{(x_i, a_i, y_i)\}_{i=1}^n$, pseudo-outcome type $m$, trade-off $\gamma \in [0,1]$, learning rates $\eta_g, \eta_\alpha$, epochs $E_1, E_2, E_3$, iterations $K$.
2: **Stage 1:** Estimate nuisance functions $\hat{\eta} = (\hat{\mu}_1, \hat{\mu}_0, \hat{\pi}_b)$; compute pseudo-outcomes $\{y_{\hat{\eta},i}^m\}$.
3: **Stage 2:** Initialize parameters $\theta$ (for $g$) and $\phi$ (for $\alpha$).
4: **for** $epoch = 1, \ldots, E_1$ **do**
5:     $\theta \leftarrow \theta - \eta_g \nabla_\theta \hat{\mathcal{L}}_{0,\alpha_\phi,\hat{\eta}}^m(g_\theta)$ {Step 1}
6: **end for**
7: **for** $iter = 1, \ldots, K$ **do**
8:     **for** $epoch = 1, \ldots, E_2$ **do**
9:        $\phi \leftarrow \phi - \eta_\alpha \nabla_\phi \hat{\mathcal{L}}_{\gamma,g_\theta,\hat{\eta}}^m(\alpha_\phi)$ {Step 2}
10:     **end for**
11:     **for** $epoch = 1, \ldots, E_3$ **do**
12:        $\theta \leftarrow \theta - \eta_g \nabla_\theta \hat{\mathcal{L}}_{\gamma,\alpha_\phi,\hat{\eta}}^m(g_\theta)$ {Step 3}
13:     **end for**
14: **end for**
15: **Output:** $g_\theta$ and $\alpha_\phi$.

---

[3]Code is available at `https://github.com/DennisFrauen/CATEForPolicy`.

# D  Details regarding simulated data

**Data-generating process.** Our general data-generating process for simulating datasets is as follows: we start by simulating initial confounders $X \sim \mathcal{U}[0, 1]$ from a uniform distribution. Then, we simulate binary treatments via

$$A \mid X \sim \text{Bernoulli}\left(\pi_b(X)\right) \tag{97}$$

for some propensity score $\pi_b(X)$. Finally, we simulate continuous outcomes via

$$Y \mid X, A \sim \mathcal{N}(\mu_A(X), \varepsilon), \tag{98}$$

where $\mu_A(X)$ denotes the response function and $\varepsilon = 0.01$ the noise level.

● **Dataset for Fig. 1.** Here, we emulate an RCT and set the propensity score to $\pi_b(X) = 0.5$. We set the response function to $\mu_a(x) = a(2\sigma(10x) - 0.5)$, where $\sigma(\cdot)$ denotes the sigmoid function. We sample a training dataset of size $n_{\text{train}} = 1000$ and a test dataset of size $n_{\text{test}} = 3000$.

● **Dataset for Fig. 2.** Here, we again emulate an RCT and set the propensity score to $\pi_b(X) = 0.5$. We set the response function to $\mu_a(x) = a(\mathbf{1}(x < -0.25)(0.6\sin(8(x + 0.25)) + 0.3) + \mathbf{1}(-0.25 < x < 0.25)(2\sigma(10(x + 2)) - 0.5) + \mathbf{1}(x > 0.25)(0.5\sin(10(x - 0.25) + 1.5))$. We sample a training dataset of size $n_{\text{train}} = 1000$ and a test dataset of size $n_{\text{test}} = 3000$.

● **Dataset for Fig. 4.** Here, we use the same propensity and response functions as in Fig. 1. We sample a training dataset of size $n_{\text{train}} = 2200$ and a test dataset of size $n_{\text{test}} = 3000$.

● **Dataset for Fig. 5.** Here, we set the propensity score to $\pi_b(X) = \sigma(0.1x)$. We then define the response function as $\mu_a(x) = a(\mathbf{1}(x < -0.25)(0.6\sin(8(x + 0.25)) + 0.3) + \mathbf{1}(-0.25 < x < 0.25)(2\sigma(10(x + 2)) - 0.5) + \mathbf{1}(x > 0.25)(0.5\sin(10(x - 0.25) + 1.5))$. We sample a training dataset of size $n_{\text{train}} = 2200$ and a test dataset of size $n_{\text{test}} = 3000$.

# E  Details regarding real-world data

The data is taken from `https://causeinfer.readthedocs.io/en/latest/data/hillstrom.html`. The dataset consists of $n = 64000$ customers who purchased a product within the last 12 months and who were involved in an email experiment: group 1 randomly received an email advertising merchandise for men, group 2 for women, and group 3 did not receive an email (control). We study the effect of receiving a men's merchandise email ($A = 1$) versus receiving no email at all ($A = 0$). Covariates $X$ include various customer features such as purchasing history. Finally, we chose $Y$ an indicator of whether people responded to the email (by clicking on the link to the website) as our outcome $Y$. We split the data into a training dataset with $50\%$ of the data, a validation set with $20\%$, and a test set with $30\%$ of the data. All details regarding our data preprocessing are provided within our codebase.[4]

---

[4]Code is available at `https://github.com/DennisFrauen/CATEForPolicy`.

# F Additional experiments

## F.1 Motivational experiments with estimated nuisance functions

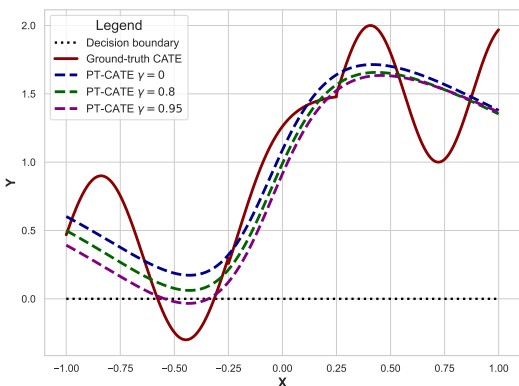

Figure 7: **Results from Figure 1 of the main paper but with using estimated nuisance functions and doubly robust pseudo-outcomes.** The dotted lines show regularized two-stage CATE estimators. The blue line corresponds to standard two-stage CATE estimation, while the green and violet lines are generated by our method for different values of $\gamma$. **The results remain robust.**

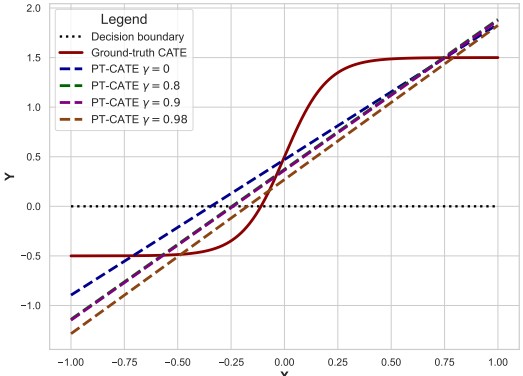

Figure 8: **Results from Figure 2 of the main paper but with using estimated nuisance functions and doubly robust pseudo-outcomes.** The dotted lines show regularized two-stage CATE estimators. The blue line corresponds to standard two-stage CATE estimation, while the other ones show the re-targeted CATE estimators using our proposed loss with different values for $\gamma$. **The results remain robust.**

## F.2 Experiments with sample splitting

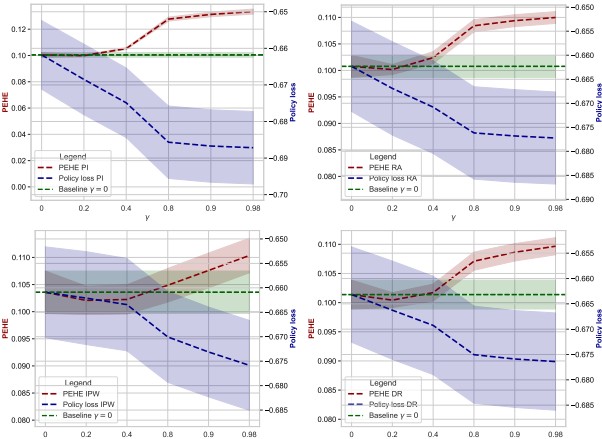

Figure 9: **Experimental results for setting A with sample splitting.** We re-ran our experiments from Fig. 4 of the main paper but use sample splitting. Shown: PEHE and policy loss over $\gamma$ (lower = better) with mean and standard errors over 5 runs. Importantly, **the results are consistent with the results of our main paper.**

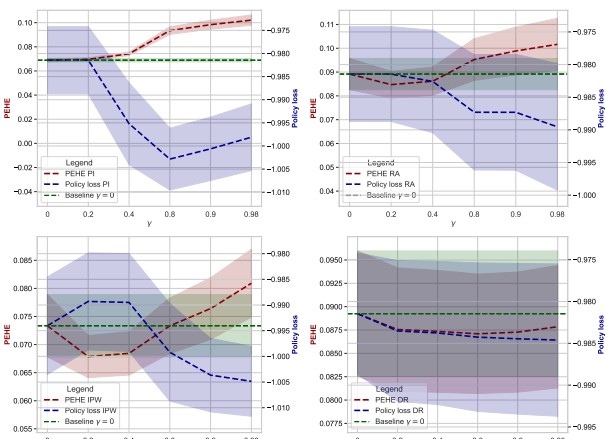

Figure 10: **Experimental results for setting B with sample splitting.** We re-ran our experiments from Fig. 5 of the main paper but use sample splitting. Shown: PEHE and policy loss over $\gamma$ (lower = better) with mean and standard errors over 5 runs. **The results are consistent with the results of our main paper.**

## F.3    Experiments with different nuisance model baselines

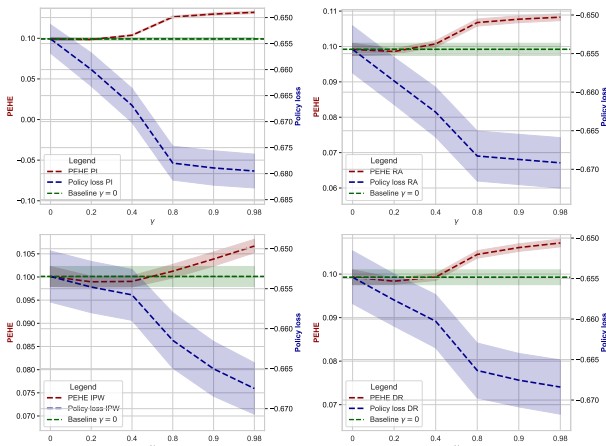

Figure 11: **Experimental results for setting A with TARNet.** We re-run our experiments from Fig. 4 of the main paper but use now TARNet (Shalit et al. 2017) for estimating the nuisance functions. Shown: PEHE and policy loss over $\gamma$ (lower = better) with mean and standard errors over 5 runs. **The results are consistent with the results of our main paper.**

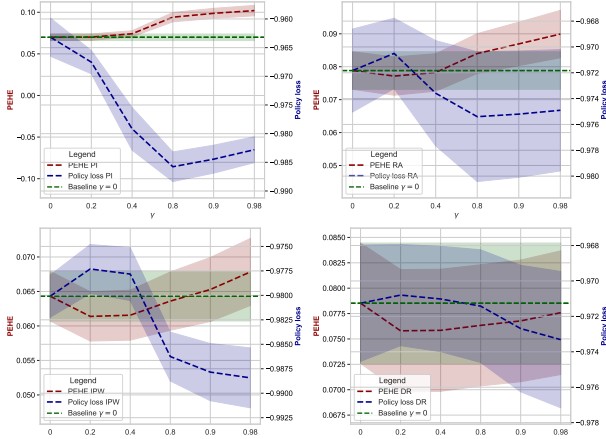

Figure 12: **Experimental results for setting B with TARNet.** We re-run our experiments from Fig. 5 of the main paper but use now TARNet (Shalit et al. 2017) for estimating the nuisance functions. Shown: PEHE and policy loss over $\gamma$ (lower = better) with mean and standard errors over 5 runs. **The results are consistent with the results of our main paper.**

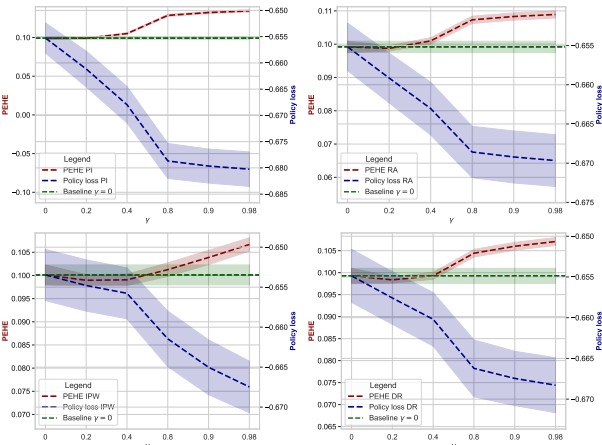

Figure 13: **Experimental results for setting A with SNet.** We re-run our experiments from Fig. 4 of the main paper but use now SNet [9] for estimating the nuisance functions. Shown: PEHE and policy loss over $\gamma$ (lower = better) with mean and standard errors over 5 runs. **The results are consistent with the results of our main paper.**

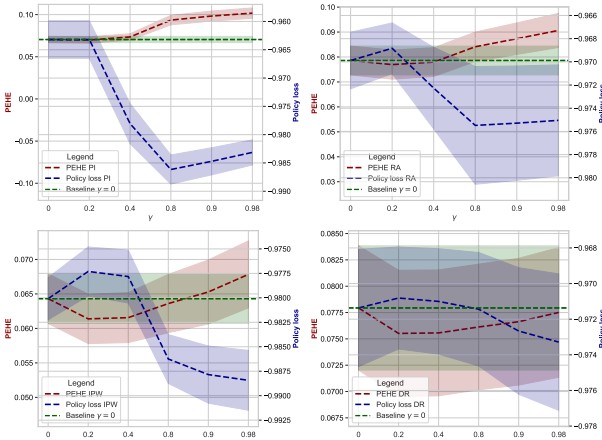

Figure 14: **Experimental results for setting B with SNet.** We re-run our experiments from Fig. 5 of the main paper but use now SNet [9] for estimating the nuisance functions. Shown: PEHE and policy loss over $\gamma$ (lower = better) with mean and standard errors over 5 runs. **The results are consistent with the results of our main paper.**

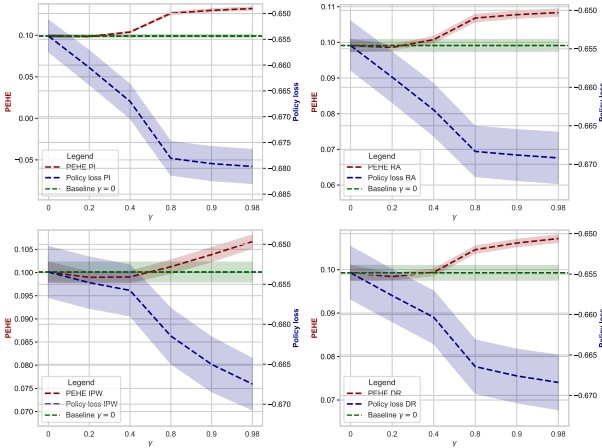

Figure 15: **Experimental results for setting A with FlexNet.** We re-run our experiments from Fig. 4 of the main paper but use now a version of FlexNet [9] for estimating the nuisance functions. Shown: PEHE and policy loss over $\gamma$ (lower = better) with mean and standard errors over 5 runs. **The results are consistent with the results of our main paper.**

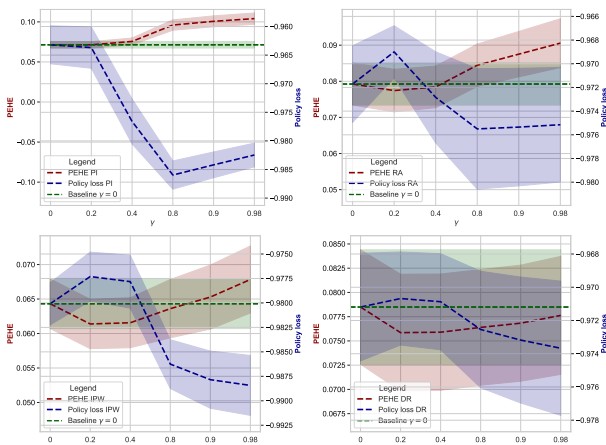

Figure 16: **Experimental results for setting B with FlexNet.** We re-run our experiments from Fig. 5 of the main paper but use now a version of FlexNet [9] for estimating the nuisance functions. Shown: PEHE and policy loss over $\gamma$ (lower = better) with mean and standard errors over 5 runs. **The results are consistent with the results of our main paper.**

## F.4 Experiments with multivariate covariates

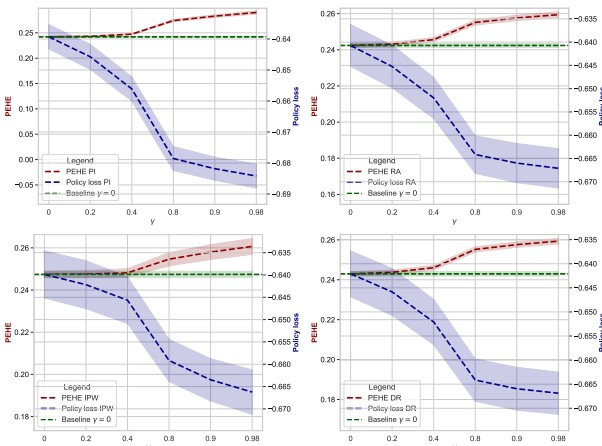

Figure 17: **Experimental results for setting A with multivariate covariates.** We re-run our experiments from Fig. 5 of the main paper but use now sample $X$ of dimension $p = 5$. Shown: PEHE and policy loss over $\gamma$ (lower = better) with mean and standard errors over 5 runs. **The results are consistent with the results of our main paper.**

