# OpenReview forum: "Treatment Effect Estimation for Optimal Decision-Making"
_NeurIPS.cc/2025/Conference — NeurIPS 2025 poster_

### Official Review · Reviewer_ytEC · 2025-06-30

**Clarity:** 4
**Significance:** 2
**Originality:** 3
**Rating:** 4
**Confidence:** 4

**Summary:**

The paper addresses a problem in causal inference: while CATE estimators are widely used to guide treatment decisions, they are typically optimized for accurate estimation — not necessarily for decision-making performance, i.e., selection of optimal treatment.  This work studies optimal decision-making based on two-stage CATE estimators. It shows that minimizing CATE estimation error does not always lead to optimal treatment policies. In particular, thresholding estimated CATEs (e.g., at zero) can yield suboptimal decisions when the true CATE is not in the function class $\mathcal{G}$ in the two-stage estimator. They propose a Policy-Targeted CATE (PT-CATE) estimator, which optimizes a combined loss function balancing CATE estimation accuracy and Policy value (i.e., decision performance). To handle the non-differentiability of the indicator function (used in policy decisions), PT-CATE uses an adaptive indicator approximation mechanism, which learns where estimation errors most impact decisions. Experiments on synthetic and real-world datasets are conducted to assess the performance of PT-CATE in terms of decision-making and estimation accuracy compared to existing two-stage CATE methods.

**Questions:**

1)	Provide more justifications about the key premise for the proposed method, i.e., the ground truth CATE is not in $\mathcal{G}$. When would practitioners choose $\mathcal{G}$ that does not contain the ground truth CATE or a close approximation of the ground truth CATE? It would be helpful to present theoretical and/or empirical results that help illustrate the impact of the distance between the ground truth CATE and $\mathcal{G}$ on the performance of the proposed approach.
2)	Provide more clarifications about the first term in the RHS of (11) in Theorem 4.4 or redo Theorem 4.4.
3)	The synthetic experiments can be improved considerably to produce results for more realistic settings that can provide more valuable insights to practitioners.

**Ethical Concerns:**

["NO or VERY MINOR ethics concerns only"]

**Final Justification:**

The authors have addressed most of my comments, so I raised my score accordingly.

**Limitations:**

There is no discussion of the limitations and/or potential negative societal impact of their work.

**Paper Formatting Concerns:**

None.

**Quality:**

3

**Strengths And Weaknesses:**

$\textbf{Strengths:}$

This works addresses an important problem: while CATE estimators are widely used to guide treatment decisions, they are typically optimized for accurate estimation — not necessarily for decision-making performance, i.e., selection of optimal treatment.  This work studies optimal decision-making based on two-stage CATE estimators. The basic idea is to minimize a weighted average of the CATE estimation error and the negative policy value of the thresholded policy where the weight $\gamma$ is a tuning parameter. Since the indicator function in the proposed loss function is non-differentiable, an adaptive indicator approximation method is used to approximate the indicator function with a adaptively weighted sigmoid function $\sigma(\alpha(x)g(x))$ where $\alpha(x)$ is learned from the data. Overall, the paper is well-written. Some theoretical results are provided to support the main claims. Experiments results are provided to support part of the main claims.


$\textbf{Weaknesses:}$

$\underline{Quality:}$ Theorem 4.4: It is unclear why the results in Theorem 4.4 imply “as long as we are able to estimate the nuisance functions $\eta$ involved in the corresponding pseudo-outcome reasonably well, we can ensure a sufficiently good second-stage learner.” The first term on the RHS of (11) depends on both $g^*$ and $\hat{g}$ both of which are included in the LHS too; this seems convoluted. Under what conditions, this first term will go to zero? Plus, the second term on the RHS of (11) suggests that in order for the proposed approach to achieve good performance, the true $\eta$ need to be estimated accurately. This also begs the question about the key premise of the proposed method, i.e., the ground-truth CATE is not in $\mathcal{G}$. If we can estimate $\eta$ well, why cannot we find a sufficiently rich function class $\mathcal{G}$ such that it contains the ground-truth CATE or a close approximation of the ground-truth CATE?

Section 4.4: The proposed approach for selecting $\gamma$ is heuristic and not data driven, and it would . Plus, the two terms on the RHS of (6) and (9) may have very different scales (e.g., one has significantly larger values than the other), so the choice of $\gamma$ is important.

Section 5: In the synthetic data experiments, the simulation settings are fairly simplistic and the results and insights gained from these experiments might not be relevant to real-world data analysis. First, in all simulations, a single X was generated and used; it would strengthen the simulations substantially to consider the case of multivariate X. Second, the true propensity score functions are also very simple for Figures 1, 2, 4 and 5. In addition, in the experiments summarized in Fig 1, Fig 2 and Fig 4, $\mathcal{G}$ is the set of linear functions. While these simulation experiments can illustrate the power of the proposed approach in a somewhat contrived set-up, they do not reflect what would be done in analysis of real data, i.e., typically a richer and more flexible $\mathcal{G}$ would be used in real data analysis. The synthetic data experiments can be improved considerably by using $\mathcal{G}$ that is often used in practice.  For Fig 5, it’s stated that “in setting B, we consider a non-linear but regularized $g$; please provide more information about the formal definition of $\mathcal{G}$ in this setting.

In the experiments using real-world data, I am unclear why it is meaningful to use the doubly robust pseudo-outcome instead, given that the ground-truth CATE is not available. It has been shown in the prior works when both working models in the doubly robust estimator are mis-specified, the resulting doubly robust estimator can be more biased than the IPW and RA/PI (e.g., Kang and Schafer, 2007, Statistical Science). In addition, how was $\mathcal{G}$ defined in the real data experiments?

To mitigate the issue of higher variance of IPW and DR, could one use stabilized PS weights in the experiments?

There is no discussion about limitations of the proposed methods.

$\underline{Clarity:}$ Line 156-158: $\eta$, $g$, and $L_\eta(g)$ should be defined or at least refer to later sections where they are defined, say $\eta$ and $g$ are defined in section 4.3.

Figure 2: Combining the left and middle panels in Figure 2 could offer a better illustration that $\alpha(x)$ detects the region in which the estimated CATE has the wrong sign.

Table 1: should the numbers in the Improv. column be multiplied by 10? In addition, it is unclear what the ‘Obs. Policy’ row represents. Should the $\gamma=0$ row be the baseline, as this row corresponds to the standard policy estimator using the estimated CATE?

$\underline{Significance:}$ Section 3.3: If the function class $\mathcal{G}$ contains the ground truth CATE, then the existing two-stage estimator $\hat{\tau}\_{\mathcal{G}}$ yields the optimal policy. It seems that distance between the ground truth CATE and the function class $\mathcal{G}$ plays a significant role in determining whether $\hat{\tau}\_{\mathcal{G}}$ can yield the optimal policy or a close-to-optimal policy. Particularly, if $\mathcal{G}$ is sufficiently rich and general, would it still be necessary to use the proposed method? Also see my earlier comment.

$\underline{Originality:}$ The ideas of using a combined loss function to balance CATE estimation error and decision performance and an adaptively weighted sigmoid function to approximate the indicator function in the loss represents incremental advances. The concern about the significance further diminishes the potential impact of this work.

---

> ### Author Rebuttal · Authors · 2025-07-29
>
> Thank you for your detailed feedback and insightful questions. Below, we respond to the specific weaknesses and questions raised in your review. We will incorporate all points marked with **Action** into the camera-ready version of our paper.
>
> ## Response to Quality
>
> **First term in the RHS of Theorem 4.4.** Thank you for your helpful comment. We agree that the original phrasing was imprecise. In Theorem 4.4. (RHS), the first term $R^m_{\gamma, \alpha, \hat{\eta}}(\hat{g}, g^\ast)$ quantifies the optimization error from the second-stage regression, while the second term $M_{\hat{\eta}, \eta}^m$  captures the estimation errors in the first-stage nuisance functions (response functions and propensity score). Thus, to ensure a small overall error, both components must be controlled. Importantly, **our result aligns with the broader literature on statistical orthogonal learning** (e.g., Foster & Syrgkanis 2023, Kennedy 2023), which decomposes the estimation error into optimization and nuisance terms. The key insight of Theorem 4.4 is how the nuisance errors affect the final estimation error, and that we obtain a doubly robust rate for the DR-pseudo outcome.
>
> **Restricting $\mathcal{G}$ when nuisances are complex** Thank you for bringing up this important point, which directly relates to the key motivation of our paper. **There are various important scenarios in which the nuisance functions can be complex, yet it is desirable to restrict the CATE model class $\mathcal{G}$.** For example:
>
> 1. **Simple propensity, complex responses:** In randomized experiments or settings with simple, well-specified propensity models, the response functions can still be complex and challenging to estimate with limited sample size, requiring restricting $\mathcal{G}$ (e.g., via regularization as in our experiments). Hence, two-stage learners, such as the IPW- or DR-learner, will only recover the best $L_2$ projection of the CATE, which can result in suboptimal decision performance, as shown in Theorem 4.1.
> 2. **Domain-specific CATE constraints (e.g., interpretability or fairness):** In many applications, one deliberately restricts $\mathcal{G}$ to satisfy domain-specific requirements (e.g., using simple models for interpretability or imposing fairness constraints). While nuisances may be estimated flexibly, the CATE model must remain constrained.
>
> **Action**: We will revise our introduction to clarify the crucial motivation for our paper.
>
> **Selecting $\gamma$.** Thank you for raising this point. We propose two practical ways of selecting an appropriate value of $\gamma$:
>
> 1. Our PT-CATE framework can be retrained for multiple values of $\gamma$, and then, the corresponding PEHE (CATE error) and policy value can be estimated (e.g., using doubly robust estimators). Now, a $\gamma$ can be selected by choosing a value that results in a large policy value improvement, while resulting in a sufficiently accurate CATE estimator. For example, in Fig. 6 in Sec. 5, a reasonable value would be $\gamma= 0.8$.
> 2. If we primarily care about decision-making, we suggest using a large value, such as $\ gamma=0.99$. This ensures that the resulting policy is comparable to applying off-policy learning (in terms of policy value), while still allowing for CATE interpretation.
>
> **Action:** We will discuss methods for selecting $\gamma$ more explicitly in the paper.
>
> **Synthetic experiments.** Thank you for your helpful suggestions regarding our synthetic experiments.  We would like to emphasize that experiments using synthetic data (usually with somewhat simplified data-generating processes) are standard in the causal inference literature. Nevertheless, we followed your suggestion and performed **additional experiments using a DGP with multivariate covariates** (p = 10) and obtained consistent results as in Fig. 4 & 5.
>
> ### PEHE (×10, mean ± std)
>
> |   $\gamma$     | 0.00        | 0.20        | 0.40        | 0.80        | 0.90        | 0.98        |
> |:-------|:------------|:------------|:------------|:------------|:------------|:------------|
> | Plugin | 2.42 ± 0.02 | 2.43 ± 0.02 | 2.47 ± 0.03 | 2.74 ± 0.05 | 2.83 ± 0.06 | 2.90 ± 0.07 |
> | RA     | 2.42 ± 0.02 | 2.43 ± 0.02 | 2.46 ± 0.03 | 2.55 ± 0.03 | 2.58 ± 0.03 | 2.59 ± 0.04 |
> | IPW    | 2.47 ± 0.03 | 2.48 ± 0.04 | 2.48 ± 0.04 | 2.55 ± 0.07 | 2.58 ± 0.08 | 2.61 ± 0.08 |
> | DR     | 2.43 ± 0.02 | 2.44 ± 0.03 | 2.46 ± 0.03 | 2.55 ± 0.03 | 2.58 ± 0.03 | 2.59 ± 0.03 |
>
> ### Policy value (×10, mean ± std)
>
> |  $\gamma$      | 0.00        | 0.20        | 0.40        | 0.80        | 0.90        | 0.98        |
> |:-------|:------------|:------------|:------------|:------------|:------------|:------------|
> | Plugin | 6.39 ± 0.09 | 6.46 ± 0.09 | 6.56 ± 0.09 | 6.79 ± 0.09 | 6.82 ± 0.08 | 6.84 ± 0.09 |
> | RA     | 6.41 ± 0.10 | 6.45 ± 0.10 | 6.52 ± 0.10 | 6.64 ± 0.09 | 6.66 ± 0.09 | 6.67 ± 0.09 |
> | IPW    | 6.40 ± 0.12 | 6.42 ± 0.12 | 6.46 ± 0.12 | 6.59 ± 0.11 | 6.63 ± 0.11 | 6.66 ± 0.11 |
> | DR     | 6.42 ± 0.11 | 6.46 ± 0.11 | 6.52 ± 0.11 | 6.64 ± 0.10 | 6.66 ± 0.10 | 6.67 ± 0.10 |
>
> Regarding the function classes $\mathcal{G}$: We chose linear function classes in Fig. 4 to showcase an important real-world use case in which we want to obtain simple interpretable CATE estimates but where the nuisances can be arbitrarily estimated. Furthermore, our additional experiments in Fig. 5 do not use a linear model class. Instead, here we use a forward neural network with 4 layers and 120 hidden neurons. To emulate model class restriction, we impose L2 regularization of 0.03. Intuitively, one can think about this setting similarly to the one displayed in Fig. 1.
>
> **DR outcomes for real-world experiments.** Thank you for your thoughtful question. First, DR pseudo-outcomes yield *asymptotically efficient* estimators when at least one nuisance function (outcome or propensity) is correctly specified. This makes them a standard choice in the literature for various causal inference problems. Furthermore, DR-outcomes are generally more robust to misspecification as only either the propensity or the response function needs to be specified correctly (doubly robustness properties). This is in contrast to RA/IPW, where misspecification of a single nuisance function can lead to several biases. While the result by Kang & Schafer cautions against DR estimators under *severe* dual misspecification, this finding is more relevant to small-sample behavior and extreme scenarios. In our setting, we have access to a sufficiently large dataset to use flexible nuisance models (neural networks) to benefit from the efficiency properties of the DR-estimator.
>
> **Stabilized IPW/ DR.** You are absolutely right: using stabilized versions of IPW/DR might further improve the estimation variance. In fact, stabilization is only one possible practical possibility for improving our method. Other methodological tweaks may be leveraged, such as propensity clipping/ trimming, or calibration of nuisance functions via isotonic regressions. However, we follow established causal inference literature and restrict our methodology and experiments to the standard versions of IPW/ DR. Our goal is to provide a clean, proof-of-concept for our policy-targeted CATE (PT-CATE) framework to evaluate the core methodology under controlled conditions and in the standard setting to ensure that the improvement stem from our proposed objective, rather than auxiliary refinements.
>
> **Action:** We will add an overview of possible methodological improvements to our paper.
>
> ## Response to Clarity
>
> Thank you for the helpful suggestions. We will update our paper accordingly.
>
> Regarding Table 1: You are correct, we multiplied the numbers by 10 for improved readability. The observational/ behavioural policy is the one generating the data, defined as $\pi(x) = \mathbb{P}(A=1 | X=x)$, (i.e., the propensity score which we estimate). In contrast, the row $\gamma= 0$ corresponds to the thresholded policy using the standard DR-learner for CATE estimation. This improves policy value compared to the observational policy, but remains inferior to our PT-CATE policies.
>
>
> ## Response to Significance
>
> To avoid redundancies, we kindly refer to our “Response to Quality” above for detailed motivational examples. In particular, there are many important applications in which $\mathcal{G}$ is restricted and does not capture the ground-truth CATE (e.g., when CATE is regularized or fairness or interpretability constraints are imposed, as in e.g., Kim and Zubizarreta et al., ICML 2023).
>
>
> ## Response to Originality
>
> Thank you for the opportunity to clarify our contributions. While combining CATE loss and policy value into a single loss may seem incremental in isolation, we believe the full set of contributions presents a non-trivial and novel advance in the field:
>
> 1. **Theoretical insight**: We formally prove that two-stage CATE estimators can be suboptimal for decision-making under thresholding (Theorem 4.1), which is a result that, to our knowledge, has not been shown before.
>
> 2. **New learning algorithm**: We propose a novel learning algorithm that iteratively corrects regions where the CATE sign is likely incorrect, using an adaptively weighted sigmoid to address optimization challenges (see Fig. 2).
>
> 3. **Mathematical guarantees**: We provide consistency and error bounds (Theorems 4.3 and 4.4), including doubly robust rates when using DR pseudo-outcomes.
>
>
> In sum, we are confident that these points present **non-trivial and novel contributions** that, to the best of our knowledge, have not been considered in previous work..
>
>
> ## Response to Questions
>
> To avoid redundancies, we kindly refer to our “Response to Clarity” above.
>
>
> ## Response to Limitations
>
> Our discussion on limitations and societal implications is included in Appendix G. We apologize for not highlighting this discussion more clearly in the main paper. We will move the discussion to the main paper of the camera-ready version in case of acceptance.

---

> > ### Comment · Reviewer_ytEC · 2025-08-04
> >
> > I appreciate the author’s effort in addressing most of my comments. My remaining concerns are as follows.
> >
> > $\textbf{First term in the RHS of Theorem 4.4.}$ I understand the second term in RHS of (11). But I do not find the response about the first term in RHS of (11) convincing, “the first term $R^m_{\gamma, \alpha, \hat{\eta}}(\hat{g}, g^\ast)$ quantifies the optimization error from the second-stage regression.” Again, $R^m_{\gamma, \alpha, \hat{\eta}}(\hat{g}, g^\ast)$ involves the same two quantities of interest, $g^\ast$ and $\hat{g}$, that are included in the LHS of (11), $||g^\ast-\hat{g}||^2$. Under what conditions would this term $R^m_{\gamma, \alpha, \hat{\eta}}(\hat{g}, g^\ast)$ go to zero? When $R^m_{\gamma, \alpha, \hat{\eta}}(\hat{g}, g^\ast)$ goes to zero under these conditions, would it be sufficient to guarantee that $||g^\ast-\hat{g}||^2$ goes to 0 too?
> >
> > $\textbf{DR outcomes for real-world experiments.}$ I am unsure that a sample size of n=64,000 is really sufficient to train deep neural networks. It may be sufficient to train shallow neural networks, which however might not be expressive enough to approximate an unknown true model accurately in real world. Also, again, how was $\mathcal{G}$ defined in the real data experiments?

---

> ### Author Response · Authors · 2025-08-05
>
> Thank you very much for acknowledging our rebuttal and for taking the time to enable a constructive discussion. Below, we respond to the two remaining points of concern.
>
>
> ## Term in Theorem 4.4.
>
> Apologies for the confusion regarding the term $R^m_{\gamma, \alpha, \hat{\eta}}(\hat{g}, g^\ast)$. Below, we provide two major arguments on **why this term is not problematic in our Theorem 4.4. and standard in the literature on orthogonal learning.**
>
>
>
> * *The term $R^m_{\gamma, \alpha, \hat{\eta}}(\hat{g}, g^\ast)$ is always non-negative in Theorem 4.4*. It holds that $R^m_{\gamma, \alpha, \hat{\eta}}(\hat{g}, g^\ast) = \mathcal{L}^m_{\gamma, \alpha, \hat{\eta}}(\hat{g}) - \mathcal{L}^m_{\gamma, \alpha, \hat{\eta}}(g^\ast) \leq 0$, because $\hat{g}$ is defined as a minimizer of $\mathcal{L}^m_{\gamma, \alpha, \hat{\eta}}(\cdot)$. Hence, we can upper bound  $R^m_{\gamma, \alpha, \hat{\eta}}(\hat{g}, g^\ast)$ with zero on the RHS in Eq. 11 and obtain a term that only depends on the nuisance estimation errors.
> * *Why do we even include the term $R^m_{\gamma, \alpha, \hat{\eta}}(\hat{g}, g^\ast)$ at all in Theorem 4.4?* The previous argument only holds if $\hat{g}$ is a perfect minimizer of the population loss $\mathcal{L}^m_{\gamma, \alpha, \hat{\eta}}(\hat{g})$. In practice, the minimum may never be achieved due to, e.g., optimization issues or finite sample errors. In this case, $R^m_{\gamma, \alpha, \hat{\eta}}(\hat{g}, g^\ast)$ may be positive and quantifies the error from the second-stage estimation. As such, we are consistent with related works on orthogonal learning (for different estimands) that either report similar second-stage error terms in population (e.g., $R_g$ in Eq. (16) in [1], or the $\mathcal{L}$ terms in Eq. 81 in [2]) or impose additional rate assumptions on the finite-sample version of this term (e.g., Definition 1b / Theorem 1 in [3]).
>
> **In summary:** The structure of Eq. 11 is consistent with established results from orthogonal learning theory; and the term $R^m_{\gamma, \alpha, \hat{\eta}}(\hat{g}, g^\ast)$ does not add additional error if $\hat{g}$ is a perfect minimizer.
>
> **The main message of Theorem 4.4. is:** The PT-CATE estimation error can be upper bounded by a term that quantifies second-stage optimization error, plus a term that only depends on the nuisance errors. In particular, the nuisance error term is of doubly robust structure for the DR pseudo-outcome.
>
> We again apologize for the ambiguity in our previous responses and are confident that the above clarifications will help to improve our theoretical section. We will include them in our paper and thank you for your valuable questions.
>
>
> ## Real-world experiments
>
> First, for the real-world experiments we only used a hidden dimension of 16 to reduce model complexity and thus risk of overfitting. Second, we would like to add that appropriate model size is a function of not only sample size, but also other factors such as the noise level within the data. As a consequence, the safest way to ensure good model generalization is to monitor the validation loss during training. During all our experiments, we always monitored both training and validation loss and adapted our architecture accordingly, **thus preventing overfitting.**
>
> You are absolutely correct in that other models may perform even better and we are sure that our model choice could be improved in practice. However, the goal of our real-world experiment is to show that PT-CATE can improve policy performance when 1) a sufficiently complex model class is used for the first stage, and 2) a constrained model class is used for the second stage. As such, we **provide empirical evidence for the main claim of our paper**, and leave further optimization of model choice to practitioners. The function class $\mathcal{G}$ is the class of linear functions similar to our synthetic experiments, thus demonstrating the effectiveness of PT-CATE in improving the policy values of linear, interpretable decision-rules.
>
> We thank you again for your valuable questions and suggestions. We are confident that incorporating responses into the main paper (particularly the additional experiments and clarifications) will improve our paper by a great deal.
>
> ## References
>
> [1] Morzywolek et al. (2024). "On Weighted Orthogonal Learners for Heterogeneous Treatment Effects". ArXiv.
>
> [2] Melnychuk et al. (2024). "Quantifying Aleatoric Uncertainty of the Treatment Effect: A Novel Orthogonal Learner". NeurIPS.
>
> [3] Foster & Syrgkanis (2023). "Orthogonal Statistical Learning". The Annals of Statistics.

---

> > ### Comment · Reviewer_ytEC · 2025-08-07
> >
> > I would like to thank the authors for providing additional clarifications about Theorem 4.4 which are very helpful. While I still have some reservation about using DR outcomes in real-world experiments, I will raise my score.

---

### Official Review · Reviewer_x8UB · 2025-07-02

**Clarity:** 2
**Significance:** 2
**Originality:** 3
**Rating:** 3
**Confidence:** 3

**Summary:**

Aiming at the decision-making performance, i.e., maximizing the outcome, the proposed method simultaneously learns the CATE regression and the classification of $I(Y_1\ge Y_0|x)$.
The proposed methods exhibit improved decision-making performance over their base learners (transformed outcome regression methods).

**Questions:**

[Q1] $α$ is said to represent uncertainty, but what does uncertainty mean in this context? For example, I don't think it refers to aleatoric uncertainty such as $0.5-|p(Y_1\ge Y_0|x)-0.5|$.

[Q2] There exist methods that estimate the uncertainty, but they require privileged information (special covariates that are available only in training). Why does uncertainty estimation benefit learners without such additional assumptions? Is $\alpha(x)$ necessary?

[Q3] Why is the alternate learning of $\phi$ and $θ$ performed a fixed number of times rather than until they converge? Is there any guarantee for this procedure?

[4] Hernández-Lobato, Daniel, et al. "Mind the nuisance: Gaussian process classification using privileged noise." Advances in Neural Information Processing Systems 27 (2014).

**Ethical Concerns:**

["NO or VERY MINOR ethics concerns only"]

**Final Justification:**

Weak baselines (base models) are resolved by adding SNet and FlexTENet.
However, these methods have strengths in their flexible architectures, and the model flexibility restriction (Settings A&B) harms their strengths.
Also, exiting work such as RMNet and OOSR (see below) seems to be closely related in that they pursue both the prediction performance and the decision-making performance, but they are not compared or discussed enough.
The authors' response to this point seems to focus on fairly subtle points that may not be of interest to most readers.

**Limitations:**

The proposed method is only applicable to transformed outcome-based methods.
The authors may want to compare with other approaches to check if this limitation is critical.

**Paper Formatting Concerns:**

No major concerns are found.

**Quality:**

2

**Strengths And Weaknesses:**

## Strengths

[S1] Causal inference aimed at decision making is an important area that has been little studied. In particular, it is important to note that when inductive bias is present, the optimal CATE estimator is not the optimal decision function.

[S2] The proposed approach of learning uncertainty simultaneously is interesting.

## Weaknesses

[W1] The experiment is limited. In particular, there is only a comparison with the basic baselines. For example, RMNet [1] proposes using the geometric mean of regression loss and classification loss, which seems to be closely related to the proposed method. There is also no comparison with architectural innovations such as FlexTENet [2] and SNet [3]. Comparison with OPE methods would also be important.

[W2] The method for estimating uncertainty is somewhat heuristic, and there is no empirical evidence regarding its necessity. See questions below.

[1] Tanimoto, Akira, et al. "Regret minimization for causal inference on large treatment space." *International Conference on Artificial Intelligence and Statistics*. PMLR, 2021.
[2] Curth, Alicia, and Mihaela Van der Schaar. "On inductive biases for heterogeneous treatment effect estimation." *Advances in Neural Information Processing Systems* 34 (2021): 15883-15894.
[3] Curth, Alicia, and Mihaela Van der Schaar. "Nonparametric estimation of heterogeneous treatment effects: From theory to learning algorithms." *International Conference on Artificial Intelligence and Statistics*. PMLR, 2021.

---

> ### Author Rebuttal · Authors · 2025-07-29
>
> Thank you for your review and for giving us the opportunity for clarification.  Below, we address your concerns in detail. In case of acceptance, we will incorporate all points marked with **Action** into the camera-ready version of our paper.
>
>
> ## Response to Weaknesses
>
> * **[W1] Baselines (first-stage learners).** Our work focuses on improving *two-stage CATE meta-learners*, such as RA-, IPW-, and DR-learners. Two-stage learners are widely considered state-of-the-art for CATE estimation [see e.g., 1] due to two properties:(i) orthogonal learners (e.g., the DR-learner) offer theoretical guarantees regarding robustness to first-stage estimation errors. We also provide similar guarantees for our PT-CATE framework (see Theorem 4.4) (ii) Two-stage learners allow for imposing constraints on the CATE directly via the function class $\mathcal{G}$, such as fairness, interpretability, or complexity control, which is often desired in real-world applications.
>
>     The baselines you mentioned (e.g., FlexTENet, and SNet) are all *first-stage learners*, i.e., model-based estimators of the response functions. Our method is designed to improve the *second stage*, i.e., the regression step that transforms plugin estimates into CATEs for decision-making. Hence, plugin learners are not directly comparable within our evaluation framework, as they do not support imposing constraints on $\mathcal{G}$, and thus do not address our research question.
>
>
>     Nonetheless, first-stage learners can be *combined* with two-stage learners by leveraging the first-stage predictions in the second-stage loss. To show that our method is robust w.r.t. the choice of first-stage learners, **we ran additional experiments in Appendix F.3** where we replaced the T-learner used in the main experiments with a TARNet. The results remain stable.
>
>
>     **Action**: To address your suggestion, we further ran **additional experiments** where we replaced the first stage with SNet and FlexTENet. **The results remain consistent and our algorithm effectively trades-off estimation error and policy value.** We will include them in the camera-ready version in case of acceptance. We summarize some of the results in the following table (we will add a plot in our revised paper, as sharing figures is prohibited during the rebuttal):
>
> ### Policy value (×10, mean ± std)
>
> |   $\gamma$                    | 0.00        | 0.20        | 0.40        | 0.80        | 0.90        | 1 *(Direct policy learning)*        |
> |:----------------------|:------------|:------------|:------------|:------------|:------------|:------------|
> | Setting A + SNet      | 6.55 ± 0.07 | 6.61 ± 0.08 | 6.68 ± 0.08 | 6.79 ± 0.08 | 6.80 ± 0.08 | 6.80 ± 0.08 |
> | Setting B + SNet      | 9.63 ± 0.08 | 9.63 ± 0.08 | 9.78 ± 0.09 | 9.86 ± 0.06 | 9.85 ± 0.06 | 9.83 ± 0.05 |
> | Setting A + FlexTENet | 6.54 ± 0.08 | 6.60 ± 0.08 | 6.67 ± 0.08 | 6.78 ± 0.08 | 6.79 ± 0.08 | 6.80 ± 0.08 |
> | Setting B + FlexTENet | 9.63 ± 0.08 | 9.64 ± 0.09 | 9.77 ± 0.10 | 9.86 ± 0.06 | 9.84 ± 0.06 | 9.82 ± 0.05 |
>
> **Baselines (OPE methods).** In our paper, we primarily focus on (widely applied) CATE-based thresholded decision rules over direct policy learning. However, our method corresponds to direct policy learning/search when setting $\gamma= 1$. As a consequence, we essentially **have direct policy learning baselines included in our experiments**, specifically the regression adjustment method [2], inverse propensity weighting (also known as importance sampling) [3], and doubly robust policy learning [4].
>
> As shown by our experiments, our PT-CATE framework effectively trades off CATE estimation error (increasing in $\gamma$) and policy value (decreasing in $\gamma$). For direct policy learning ($\gamma= 1$), the policy value is the largest, as it is the only objective being optimized. However, CATE performance is the worst, and the resulting thresholding function is not interpretable as a CATE anymore. This is particularly relevant in various disciplines (e.g., medicine), where CATE-based decision-making is important due to its interpretability, and potentially negative side effects need to be balanced.
> **Action:** We will add a clarification to the paper.
>
> **Novelty as compared to RMNet.** Thank you for bringing this paper to our attention. RMNet targets a fundamentally different setting: it is designed for a multi-action causal inference setting. In contrast, our setting considers the standard setting with binary treatments and emphasizes interpretable, thresholded policies derived from two-stage learners. Importantly, RMNet is a *first-stage learner*, while ours is a *second-stage learner*. Furthermore, unlike RMNet, our contributions are: (i) A mathematical proof of the suboptimality of standard two-stage CATE learners for decision-making under constraints (Theorem 4.1); (ii) a novel learning objective that trades off CATE accuracy and policy value in the second stage; and (iii) a principled learning algorithm with guarantees under nuisance estimation error (Theorem 4.4) and empirical validation with several nuisance models. To the best of our knowledge, **none of these contributions are addressed by RMNet or any existing related work.**
> **Action:** We will add this discussion to our related work.
>
> * **[W2] Uncertainty.** Here, we kindly refer you to our response to Q1 below.
>
>
> ## Response to Questions
>
> * **Q1: Interpretation of $\alpha$ as uncertainty.** Thank you for bringing up this point. We would like to clarify that the purpose of $\alpha$ is **not** quantifying estimation uncertainty nor uncertainty of the data-generating process (i.e., epistemic or aleatoric uncertainty). Instead, $\alpha$ ensures that the resulting PC-CATE policy is a **stochastic policy** that primarily ensures the **feasibility of our optimization procedure.**
>
>     Stochastic policies are common in large parts of the off-policy learning (OPL) and reinforcement learning literature (RL) (e.g., as in [5] or [6]). Even though the ground-truth policy is often deterministic, learning algorithms in OPL or RL often minimize objectives over a class of stochastic policies, as this brings benefits for, e.g., gradient-based optimization by introducing smoothness.
>
>     **Intuition in our setting:** Let us consider Figure 4.1. (left). Here, we would like to classify the covariate space at which the red and the blue lines differ in their sign. In other words, we would like to learn the deterministic function $f(x) = I(\hat{\tau}_\mathcal{G} \tau(x) &lt; 0)$. One way to think about our approach is to approximate the (non-differentiable function $f(x) \approx \sigma(h(X))$ with a smooth classifier, where $h(X)$ is some real-valued score. This classifier is a smooth function that can be trained with stochastic gradient descent. In our setting, we decompose $h(X) = \alpha(X) g(X)$, where $g(X)$ should remain interpretable as CATE, and $\alpha(X) > 0$ determines the “steepness” of the sigmoid function. This is what we refer to as “uncertainty”; however, this only refers to the stochasticity of the learned policy, **not** to quantified epistemic or aleatoric uncertainty.
>
>     **Action:** We admit that the notion of “uncertainty” is ambiguous in our setting. We will refrain from using uncertainty and highlight differences between stochastic policies in our paper.
>
> * **Q2: Methods for uncertainty quantification.** We kindly ask you to refer to our above response on why our current approach is unrelated to methods for epistemic or aleatoric uncertainty quantification. However, we agree that incorporating such methods into our approach may be an interesting future direction. For example, one could optimize for distributional versions of the policy value or combine our method with Bayesian estimation or conformal prediction. This would, however, be **unrelated to $\alpha(X)$ in our setup**, which primarily enables gradient-based optimization.
>
>     **Action:** We will add these considerations to our outlook on future work.
>
> * **Q3: Alternating optimization.** Thank you for your thoughtful question. In principle, our alternating learning procedure can indeed be iterated over arbitrary rounds. However, we observed convergence already after a single iteration across all our experiments. This is due to the structure of our algorithm: the initial CATE estimation provides a reasonable approximation of the treatment effect; the subsequent fitting of $\alpha$ identifies regions where decision errors (i.e., incorrect signs) occur; and the final refitting step corrects for those errors, resulting in improved downstream decision-making. We emphasize that further iterations can be applied if needed, as our framework does not preclude this. Yet, our empirical results demonstrate that a single iteration suffices to achieve substantial improvements in policy value and CATE estimation across various settings.
>
>     **Theoretical guarantees:** Alternate optimization is a standard approach in min-max and adversarial learning, and has also been used in causal inference literature [6]. While there exist works that provide convergence guarantees for specific alternate optimization setups [7], we view formal convergence guarantees of alternating optimization beyond the scope of our current contribution.
>
> ## References
>
> [1] Kennedy et al. Minimax rates for heterogeneous causal effect estimation. Annals of Statistics.
>
> [2] Qian & Murphy (2011). Performance guarantees for individualized treatment rules. Annals of Statistics.
>
> [3] Swaminathan and Joachims (2015). Counterfactual risk minimization: Learning from logged bandit feedback. ICML 2015.
>
> [4] Athey and Wager (2021). Policy learning with observational data. Econometrica.
>
> [5] Kallus. Balanced policy evaluation and learning. NeurIPS 2018.
>
> [6] Schweisthal et al. Reliable Off-Policy Learning for Dosage Combinations. NeurIPS 2023.
>
> [7] Lin et al. On Gradient Descent Ascent for Nonconvex-Concave Minimax Problems. ICML 2022.

---

> ### Comment · Reviewer_x8UB · 2025-08-05
>
> I appreciate that adding SOTA base learners has improved the reliability of the results, and I have slightly increased my score accordingly.
> However, those SOTA methods mainly gain performance through model flexibility, whereas Settings A and B (if I understand correctly) intentionally impose strong restrictions on model complexity, which is not a fair comparison.
> Or, if such flexibility constraints are a very important part of the problem setting, then the paper title should be changed accordingly.
>
> Regarding [W1], if the goal is to jointly pursue causal estimation accuracy and decision performance, I believe the paper should compare with related work such as RMNet or, although not exactly the same objective, OOSR [1], either theoretically or empirically.
> If, on the other hand, the contribution is limited to improving only the second stage learners, the scope looks rather narrow and does not match top venue standards, in my view; my overall evaluation thus remains closer to reject.
>
> Additionally, the authors classify RMNet as a first-stage learner. Yet RMNet performs a classification learning w.r.t. a pretrained baseline model, which to me looks like a second-stage learner. What precise definition of second-stage learner are the authors using, and why is focusing on that subclass important for the community?
>
> [1] Zou, H., et al. “Counterfactual Prediction for Outcome-Oriented Treatments.” ICML 2022.

---

> ### Author Response · Authors · 2025-08-05
>
> Thank you very much for acknowledging our rebuttal and for raising your score. Please allow us to elaborate on the remaining points of concerns.
>
> **Setting of the paper:** The central research questions of our paper concern the optimality of (i) two-stage metal-learners for decision making under (ii) constrained second-stage models. Below, we provide arguments why both (i) and (ii) are highly relevant in both state-off-the-art causal inference research and practical applications.
>
> (i) **Why “only”  two-stage learners?** Two-stage CATE learners are widely considered to be state-off-the-art, both empirically [1] and theoretically [2]. Important two-stage learners such as the DR-learners are motivated by semiparametric efficiency theory, which is *the theory at the core of causal inference for the last two decades* [see e.g., 3, 4, 5]. As such, two-stage learners constitute a major approach to CATE estimation and are highly relevant in both research and practical applications.
>
> (ii) **Why consider “restricted” second-stage model classes $\mathcal{G}$?** One key advantage of two-stage learners is that they allow to specify a model class for the CATE directly. In particular, this allows customized model specification (i.e., “constraints”) of $\mathcal{G}$ to incorporate various application specific desiderata. **Importantly, these “constraints” do not imply that our methodology is restricted to a niche subset of methodology.** On the contrary, the following use cases are crucial for many practical applications:
>
> * *Interpretability constraints:* In sensitive applications such as medicine, physicians need inherently simple models to justify high-stakes decision-making, potentially deciding about life or death of a patient. Here, linear models or simple decision-trees are commonly deployed. Works in causal inference that consider interpretable models include [6].
> * *Fairness constraints:* In many applications, fairness constraints need to be satisfied by a decision-models. **Fair decision-making is a large and established field** in the machine learning community, and previous work has explored incorporation fairness into causal inference [e.g., 7, 8, 9]
> * *Privacy constraints:* In many sensitive applications, we have to ensure that our CATE predictions cannot be used to make conclusions about individual data-points. Works on two-stage learners that consider privacy include [10, 11].
>
> **All the above applications are special cases of constrained model classes that fall within our framework, and our PT-CATE framework is readily applicable to enhance decision-making based on interpretable, fair, or private CATE models.**
>
> To the best of our knowledge, **our work is the first that** (i) theoretically analyzes the gap in decision-making performance of such constrained two-stage learners, and (ii) proposes methodology to address this gap. Both papers you mentioned (OOSR and RMNet) aim at improving the overall policy value of the CATE learner and do not target constrained two-stage learners. While RMNet can be interpreted as using a regression-adjustment first-stage model, neither theory nor methodology is intended for our setting. Additionally, RMNet is restricted to regression-adjustment, while our methodology is compatible with established two-stage learners such as the DR-learner, for which we also provide doubly-robust rates (Theorem 4.4.).
>
> References
>
> [1] Curth et al. (2021). Nonparametric estimation of heterogeneous treatment effects: From theory to learning algorithms. AISTATS.
>
> [2] Kennedy et al. (2023). Minimax rates for heterogeneous causal effect estimation. Annals of Statistics
>
> [3] Van der Laan. (2006). Targeted Maximum Likelihood Learning. The International Journal of Biostatistics.
>
> [4] Chernozhukov et al. (2018) Double/De-Biased Machine Learning for Treatment and Causal Parameters. Econometrica.
>
> [5] Foster et al. (2023). Orthogonal statistical learning. Annals of Statistics.
>
> [6] Tschernutter et a. (2022). Interpretable Off-Policy Learning via Hyperbox Search. ICML.
>
> [7] Fang et al. (2021). Fairness-Oriented Learning for Optimal Individualized Treatment Rules. Journal of the American Statistical Association.
>
> [8] Kim et al. (2023). Fair and Robust Estimation of Heterogeneous Treatment Effects for Policy Learning. ICML.
>
> [9] Frauen et al. (2024). Fair off-policy learning from observational data. ICML
>
> [10] Niu et al. (2022). Differentially Private Estimation of Heterogeneous Causal Effects. CLeaR.
>
> [11] Schroeder et al. (2025). PrivATE: Differentially Private Confidence Intervals for Average Treatment Effects. ICLR.

---

> > ### Comment · Reviewer_x8UB · 2025-08-06
> >
> > Thank you for your response. Since “two-stage learner” has not been formally defined, I do not see why RMNet is classified as a first-stage learner; RMNet itself also appears to involve two learning stages.
> > Moreover, in addition to the motivation, the loss functions look very similar, differing only in a minor point of using a geometric or an arithmetic mean.
> >
> > The theoretical work the authors mentioned mainly concerns double robustness, while only the DR-learner is actually doubly robust among the base learners examined in the paper.
> > It is therefore unclear why this theory is presented as evidence for the advantages of all two-stage learners.
> > Empirically as well, non-two-stage methods such as FlexTENet often achieve superior performance.
> >
> > Finally, while I understand that limiting model complexity can be important in certain applications, this is not always the case; neither the title nor the abstract conveys that such restrictions are central in the problem setting of this paper.
> >
> > For these reasons, I would like to maintain my current score.

---

> > > ### Author Response · Authors · 2025-08-06
> > >
> > > Thank you for your response and additional feedback. If we may, we would like to clarify a few key points, as we believe certain aspects of our work may not have been communicated as clearly as intended.
> > >
> > > **Regarding “limited model complexity”.** Thank you. We acknowledge that our title and abstract may not be written sufficiently precisely to accurately reflect the full scope of our paper correctly. We will take your feedback at heart and modify them accordingly, in particular, we will clearly state and motivate why the key research question is on enhancing two-stage learners with constrained second-stage models.
> > >
> > > **Two stage learners and differences to RMNet.** The formal definition of two-stage learners in our setting is as follows [1]: First, nuisance models are fitted to estimate (a subset of) the nuisance functions $\eta = (\pi, \mu_1, \mu_0)$, where $\pi$ is the propensity score and $\mu_a(x) = E[Y | X=x , A=a]$ are the response functions. Then, in a second-sage loss $\mathcal{L}_{{\eta}}(g)$, is minimized over a function class $\mathcal{G}$. Importantly, **$\tau$ is directly obtained by a minimization over $\mathcal{G}$, and *not* in a plugin fashion via $\tau(x) = \mu_1(x) - \mu_0(x)$.**
> > >
> > > In RMNet, the authors propose to minimize the geometric mean of the MSE and a ranking loss. For the MSE, no nuisance models $\eta$ are trained (i.e., no first stage). For the ranking loss, the authors fit a first stage model to estimate $v(x) = E[Y | X=x]$, which does not coincide with any of the nuisances above. Once RMNet estimates the response functions $\mu_a(x) = E[Y | X=x , A=a]$, the CATE is obtained via plugin fashion $\tau(x) = \mu_1(x) - \mu_0(x)$ and not as the minimizer to a second-stage loss. As a consequence:
> > > * **The loss can not be written as a two-stage learner for CATE estimation, RMSNet is fundamentally a first stage estimator.**
> > > * It is not clear what population target the RMNet loss minimizes as their loss is only an upper bound for the MSE. Once we restrict the function class $\mathcal{G}$, **RMNet would not provide any meaningful projection on the CATE** (in contrast to our PT-CATE framework).
> > >
> > > This is not to say that our approach is generally “superior” to the RMNet one, just that **both approaches target fundamentally different settings:** RMNet aims to improve first-stage decision-making performance, particularly in high-dimensional treatment settings. In contrast, our aim is to improve decision-making of two-stage learners when intentionally restricting the second-stage model class.
> > >
> > > **Theoretical contributions.** Thank you for pointing this out, we will add a clarification to the paper. You are correct that double robustness is limited to the DR-learner. However, we would like to note that this is standard in established literature [1] and to be expected. Furthermore, the possibility of incorporating constraints on $\mathcal{G}$ is shared across all two-stage learners, and so is our theoretical result regarding decision-optimality (Theorem 4.1.).
> > >
> > > We acknowledge that in certain situations, first-stage learners (such as directly using FlexTENet for CATE estimation) can outperform two-stage learners (e.g., in small sample regimes or settings with high CATE complexity). However, first-stage learners are fundamentally unable to incorporate direct constraints on the estimated CATE by restricting $\mathcal{G}$, and are thus not in the scope of our paper.
> > >
> > > Thank you again for your continuous responses during the discussion period. We sincerely appreciate your time and effort in reviewing our work.
> > >
> > >
> > >
> > > References
> > >
> > > [1] Curth et al. (2021). Nonparametric estimation of heterogeneous treatment effects: From theory to learning algorithms. AISTATS.

---

### Official Review · Reviewer_XmcU · 2025-07-02

**Clarity:** 3
**Significance:** 3
**Originality:** 3
**Rating:** 5
**Confidence:** 5

**Summary:**

This paper investigates a critical disconnect between the goals of estimating CATE and using those estimates for optimal decision-making. The authors argue that state-of-the-art CATE estimators, such as the DR-learner, are optimized to minimize estimation error across the entire covariate space. However, for decision-making (e.g., assigning treatment if CATE > 0), accuracy is most crucial near the decision boundary. This misalignment can lead to suboptimal policies, especially when the model class for the CATE estimator is restricted (e.g., by regularization or for interpretability).

The authors propose a novel learning objective that balances the CATE estimation error with the policy value through a tunable hyperparameter. To overcome the challenge of optimizing the non-differentiable policy value term, they introduce an adaptively-smoothed approximation.

The authors provide theoretical proofs for the suboptimality of standard CATE estimators in this context and establish consistency and error rate guarantees for their proposed method. Through experiments on both synthetic and real-world datasets, they demonstrate that their PT-CATE approach can successfully trade a small amount of CATE estimation accuracy for a significant improvement in policy value.

**Questions:**

See weakness.

**Ethical Concerns:**

["NO or VERY MINOR ethics concerns only"]

**Final Justification:**

I increased the score to 5.

**Limitations:**

Yes

**Quality:**

3

**Strengths And Weaknesses:**

## Strengths

- The paper addresses a subtle but highly practical issue. The distinction between optimizing for estimation versus decision-making is a crucial insight, and the authors do an excellent job of motivating it with clear, intuitive examples and visualizations.

- The proposed PT-CATE objective is an elegant solution to the identified problem. The method for handling the non-differentiable indicator function via an adaptive smoothing function ($\alpha(X)$) is innovative and allows for gradient-based optimization. The approach is well-grounded in theory, with formal proofs backing its main claims.

- The paper provides comprehensive experiments.

## Weaknesses

- The method's key hyper-parameter, $\gamma$, controls the trade-off between estimation accuracy and decision value. I would suggest authors to give some data-driven heuristic approaches for practical guidance.

- The method is benefit from when the CATE model class is mis-specified or restricted ($\tau\notin G$). The paper acknowledges that if the model class is flexible enough to perfectly capture the true CATE, the proposed method will not offer an improvement over standard learners. In practice, we often use simpler model due to the limited sample size or better interpretation. As noted by [1], a mis-specified model can result in near-optimal policy. Is there a way to balance the model complexity, estimation accuracy and policy value?


[1] Besbes, O., Phillips, R., & Zeevi, A. (2010). Testing the validity of a demand model: An operations perspective. Manufacturing & Service Operations Management, 12(1), 162-183.

---

> ### Author Rebuttal · Authors · 2025-07-29
>
> Thank you very much for your thoughtful review! Below, we address all your comments and suggestions. We will incorporate all points marked with **Action** into the camera-ready version of our paper.
>
> ## Response to Weaknesses
>
> 1. **Selecting $\gamma$.** Thank you for raising this critical point. We propose two different ways of selecting an appropriate value of $\gamma$
>     - Our PT-CATE framework can be retrained for multiple values of $\gamma$, and then, the corresponding PEHE (CATE error) and policy value can be estimated (e.g., using doubly robust estimators). Now, a $\gamma$ can be selected similar to an Elbow plot (e.g., for selecting the optimal number of clusters in a clustering algorithm): choose a value that results in a large policy value improvement, while resulting in a sufficiently accurate CATE estimator. For example, consider our real-world experiments (Fig. 6 in Sec. 5). Here, a reasonable value would be $\gamma=0.8$, which is desirable in terms of both policy value improvement and CATE accuracy.
>     - If we primarily care about decision-making, we suggest using a large value, such as $\gamma=0.99$. This ensures that the resulting policy is comparable to applying off-policy learning (in terms of policy value), while still allowing for CATE interpretation. In contrast, the decision function for off-policy learning (i.e., $\gamma=1$) has no interpretation as a treatment effect. The exact value (e.g., $\gamma\in\{0.98,0.99\}$) can be determined via the same approach as in bullet point a.
>     **Action:** We will discuss methods for selecting $\gamma$ more explicitly in the paper.
>
>
>
> 1. **Model complexity, estimation accuracy, and policy value.** Thank you for this interesting question. In general, the discrepancy between CATE-based and optimal policy value is determined by how well the CATE projection using the model class $\mathcal{G}$ can estimate the sign of the ground-truth CATE $\tau$. This depends on several factors, including (i) the error of the optimal projection (related to model complexity), and (ii) the structure of the ground-truth CATE $\tau$.
>
>     As an illustration, consider Figure 2 (left) from our paper, where the discrepancy in policy value is characterized by the distance of the two points where the blue (estimator) and red line (ground-truth CATE) intersect zero. For (i), we could make the red line steeper (increasing the projection error), and thus can arbitrarily widen this gap to obtain an arbitrary discrepancy between CATE-based and optimal policy value. For (ii), we could shift the red line upwards, which would also widen the gap without changing the projection error.
>
>
>     If the projection error is small (i.e., our learned CATE is close to the ground-truth CATE, we are able to bound the discrepancy in policy value as follows.
>
>     **Lemma.** Let let the true CATE be $\tau(x)=\mu_{1}(x)-\mu_{0}(x)\in L^{2}(\mathbb{P})$. For a function class $\mathcal{G}\subset L^{2}(\mathbb{P})$, define its $L^{2}$‐projection of $\tau$ as $g^{\ast} = \arg\min_{g\in\mathcal{G}} \mathbb{E} \left[(\tau(X)-g(X))^{2}\right]$ and $\epsilon_2 =\|\tau-g^{\ast}\|_{L^2}$.
>
>     Furthermore, we introduce the corresponding thresholded treatment policies $\pi_{\tau}(x)=1(\tau(x)>0)$, $\pi_{g^{\ast}}(x)=\mathbf{1}(g^{\star}(x)>0)$ and the (shifted) policy value $V(\pi) = \mathbb{E} \bigl[\pi(X) \tau(X)\bigr]$. Then, it holds that $0 \leq V \bigl(\pi_{\tau}\bigr)-V \bigl(\pi_{g^{\star}}\bigr) \leq \epsilon_{2}$.
>
>     **Proof (sketch).** We define the *disagreement set* $\mathcal{M}:=\{ x | sign(\tau(x)) \neq sign(g^{\star}(x)) \}$. Then, it holds that
> $\Delta =V(\pi_{\tau})-V(\pi_{g^{\star}}) =\mathbb{E}\bigl[\tau(X)\bigl(\pi_{\tau}(X)-\pi_{g^{\star}}(X)\bigr)\bigr]
> =\mathbb{E}\bigl[\tau(X)\mathbf{1}_{\mathcal{M}}(X)\bigr]$. On $\mathcal{M}$ we have $\tau(X)g^{\star}(X)\le 0$, hence
> $|\tau(X)|\le|\tau(X)-g^{\star}(X)|$.
>
> This implies $\Delta \leq \mathbb{E}\bigl[|\tau(X)-g^{\star}(X)|1_{\mathcal{M}}(X)\bigr] \leq \mathbb{E}\bigl[|\tau(X)-g^{\star}(X)|\bigr] =|\tau-g^{\star}|_{1} \leq \epsilon_2$., where the last inequality follows from Hölder’s inequality.
>
> Note, however, that in many applications the projection error will be large as a result of constraints on the function class $\mathcal{G}$ to ensure, e.g., interpretability [1] or fairness [2, 3], or to accommodate small data regimes. In such scenarios, the discrepancy in policy values may be large, and our method would provide substantial benefits over classical two-stage learners.
>
> **Action:** We will add the new theoretical result to our paper. We will also add the paper mentioned to our related work, noting, however, that it does not focus on a causal inference setting or CATE estimation directly.
>
> ## References
>
> [1] Tschernutter et al. Interpretable Off-Policy Learning via Hyperbox Search. ICML 2022.
>
> [2] Fang et al. (2021). Fairness-Oriented Learning for Optimal Individualized Treatment Rules. Journal of the American Statistical Association.
>
> [3] Kim et al. Fair and Robust Estimation of Heterogeneous Treatment Effects for Policy Learning. ICML 2023.

---

> > ### Comment · Area_Chair_Dz24 · 2025-08-05
> >
> > Dear Reviewer XmcU,
> >
> > Please respond to the authors. Thanks.
> >
> > Best,
> > AC

---

> > ### Comment · Reviewer_XmcU · 2025-08-06
> >
> > I thank authors for detailed responses on the two weakness points. The second point is well addressed. I would recommend authors to discuss the first point with the second. How can you select $\gamma$ with pre-selected function class $\mathcal{G}$?

---

### Official Review · Reviewer_9gJM · 2025-07-05

**Clarity:** 2
**Significance:** 3
**Originality:** 3
**Rating:** 5
**Confidence:** 3

**Summary:**

This paper investigates how commonly used two-stage CATE estimators (like DR-learner), though accurate for estimating treatment effects, can be suboptimal for decision-making. The key insight is that these estimators often focus on regions far from the decision boundary, which doesn't help practical decisions. To address this, the authors propose a new two-stage learning objective that balances CATE estimation with decision performance. They develop a neural method to optimize this objective and demonstrate its effectiveness both theoretically and empirically. This is the first work to adapt state-of-the-art CATE methods specifically for decision-making.

**Questions:**

1. This proposed policy-targeted CATE introduced a tuning parameter \gamma. In the paper, this parameter is proposed to to selected on domain knowledge. In practice, is there any valid way to better select it? How sensitive does it affect the performance of the estimated policy? More experiment results to justify the robustness of the method is helpful.

**Ethical Concerns:**

["NO or VERY MINOR ethics concerns only"]

**Final Justification:**

I will keep my positive rating after reading author's rebuttal

**Limitations:**

1. Though the paper provides simulation study to compare with standard two-stage CATE learners, there seems no comparison with direct policy search using OPE. Will the proposed method outperform these methods?
2. A minor one: the research problem in this paper is novel but this is not reflected in the title of this paper. Probability adding more general description in the title? For now it seems it is too broad.

**Quality:**

3

**Strengths And Weaknesses:**

1. The paper provide s a novel method to mitigate the gap between CATE estimators and the optimal decision policy.
2. The paper provides solid theoretical results to justify the proposed research questions. This is also validated in simulation analysis.

---

> ### Author Rebuttal · Authors · 2025-07-29
>
> Thank you very much for your positive review! Below, we address your comments and suggestions in detail. We will incorporate all points marked with **Action** into the camera-ready version of our paper.
>
>
> ## Response to Questions
>
>
>
> 1. **Selecting $\gamma$.** Thank you for raising this critical point. We propose two different ways of selecting an appropriate value of $\gamma$
>     - Our PT-CATE framework can be retrained for multiple values of $\gamma$, and, then, the corresponding PEHE (CATE error) and policy value can be estimated (e.g., using doubly robust estimators). Now, a $\gamma$ can be selected similar to an Elbow plot (e.g., for selecting the optimal number of clusters in a clustering algorithm): choose a value that results in a large policy value improvement, while resulting in a sufficiently accurate CATE estimator. For example, consider our real-world experiments (Fig. 6 in Sec. 5). Here, a reasonable value would be $\gamma=0.8$, which is desirable in terms of both policy value improvement and CATE accuracy.
>     - If we primarily care about decision-making, we suggest using a large value, such as $\gamma=0.99$. This ensures that the resulting policy is comparable to applying off-policy learning (in terms of policy value), while still allowing for CATE interpretation. In contrast, the decision function for off-policy learning (i.e., $\gamma=1$) has no interpretation as a treatment effect. The exact value (e.g., $\gamma \in \{0.98, 0.99\}$) can be determined via the same approach as in bullet point A.
>
>     **Action:** We will discuss methods for selecting $\gamma$ more explicitly in the paper.
>
>
> ## Response to Limitations
>
>
>
> 1. **Comparison to direct policy learning.** In our paper, we primarily focus on (widely applied) CATE-based thresholded decision rules over direct policy learning. However, it turns out that our method corresponds to direct policy learning/search when setting $\gamma= 1$. As a consequence, we essentially have **direct policy learning baselines included in our experiments**, specifically the regression adjustment method [1], inverse propensity weighting (also known as importance sampling) [2], and doubly robust policy learning [3].
>
>     As indicated by our experiments, our PT-CATE framework effectively trades off the CATE estimation error (which increases with $\gamma$) and the policy value (which decreases with $\gamma$). For direct policy learning ($\gamma= 1$), the policy value is the largest, as it is the only objective being optimized. However, CATE performance is the worst, and the resulting thresholding function is no longer interpretable as a CATE. This is particularly relevant in various disciplines (e.g., medicine), where CATE-based decision-making is essential due to its interpretability, and potentially harmful side effects need to be balanced.
>
>
>     **Action:** We will add a clarification to the paper.
>
> 1. **Paper title.** Thank you for your suggestion. We are open to renaming the paper after the rebuttal period (which is in line with the guidelines of NeurIPS). A possible title would be “Re-targeting heterogeneous treatment effects for optimal decision-making/ policy learning”. We are happy to consider any other suggestions you may have.
>
>
> ## References
>
> [1] Qian & Murphy (2011). Performance guarantees for individualized treatment rules. Annals of Statistics.
>
> [2] Swaminathan and Joachims (2015). Counterfactual risk minimization: Learning from logged bandit feedback. ICML 2015.
>
> [3] Athey and Wager (2021). Policy learning with observational data. Econometrica.

---

> > ### Comment · Reviewer_9gJM · 2025-08-03
> >
> > thanks for the rebuttal. I acknowledge reading the rebuttal and keep my score.

---

> > > ### Author Response · Authors · 2025-08-04
> > >
> > > Thank you for acknowledging our rebuttal and your positive score! We will make sure to include your suggestions in the paper.

---

### Official Review · Reviewer_Yh69 · 2025-07-06

**Clarity:** 3
**Significance:** 3
**Originality:** 3
**Rating:** 5
**Confidence:** 4

**Summary:**

The paper revisits the crucial problem that there is a difference between optimizing CATE estimation error and actual decision-making \when using two-stage CATE estimators (e.g., DR-learner). Interestingly, the paper proves theoretically that minimizing CATE estimation error alone can yield suboptimal treatment policies. Finally, the authors propose a novel γ-policy-targeted CATE objective, which balances CATE estimation accuracy with policy value optimization by introducing a trade-off hyperparameter $\gamma$.

**Questions:**

1. Is it possible to show on what occasions CATE estimation will give a precise causal decision-making? It is believed that if the CATE estimator perfectly equals the true CATE, i.e., $\hat{\tau}=\tau^*$, the causal decision making would be perfect. Is it possible to relax this condition? I guess Theorem 4.3 and Theorem 4.4 might be helpful to this question. I understand this may not be the main target of this paper, just out of curiosity, and your response to this question will not affect my score.

**Ethical Concerns:**

["NO or VERY MINOR ethics concerns only"]

**Final Justification:**

I maintain the score 5.

**Limitations:**

See above.

**Paper Formatting Concerns:**

Good.

**Quality:**

3

**Strengths And Weaknesses:**

Strengths:

1. This paper targets a very important research question: the gap between CATE estimation and causal decision making.
2. This paper gives a theoretical explanation for the suboptimality of the two-stage CATE estimator in causal decision making.
3. This paper proposes a very interesting and novel learning objective for retargeting CATE.

Weaknesses:

1. I think this paper uses too much content to state the gap between CATE estimation and causal decision making, which is actually not the novel point of this paper. I understand that maybe readers may not be very familiar with the literature, but I suggest the authors move these parts to supplementary or directly suggest refer [1]. I prefer to see more theoretical explanations and analysis on why the CATE estimator might lead to suboptimality in decision making. This is not provided in [1], so it would be helpful to strengthen your contribution, while you just leave Theorem 4.1 with a short interpretation.

2. I found the new learning objective $\gamma$-policy-targeted CATE objective is not very intuitive. It indeed balances CATE estimation accuracy and policy learning, but how can we control it theoretically? The authors provide some empirical insights on $\gamma$ selection, but is it possible to give a specific solution? What is the definition of optimal $\gamma$?


[1] Causal Decision Making and Causal Effect Estimation Are Not the Same…and Why It Matters.

---

> ### Author Rebuttal · Authors · 2025-07-29
>
> Thank you for your positive review and your helpful comments. Below we have drafted careful responses to your suggestions. We will incorporate all points marked with **Action** into the camera-ready version of our paper.
>
> ## Response to Weaknesses
>
> 1. **Emphasis on CATE estimation vs decision-making.** Thank you for pointing this out. **Action:** We will follow your suggestions closely, and we will rewrite our introduction to put less emphasis on the general topic of “CATE estimation vs decision-making”. Instead, we will elaborate more on the specific research question we target in our paper: the decision-making capabilities of two-stage learners, and a novel two-stage learning algorithm to improve these.
>
>     **Further analysis on the suboptimality of CATE estimators.** We are happy to provide such analysis and kindly refer you to our response to “Questions” below.
>
> 1. **Selection of $\gamma$.** Thank you for raising this point. In general, there is no “mathematically optimal” way of choosing $\gamma$ as $\gamma$ defines the trade-off between CATE estimation performance and policy value we wish to achieve. In practice, we propose two different ways of selecting an appropriate value of $\gamma$:
>
>     - Our PT-CATE framework can be retrained for multiple values of $\gamma$, and then, the corresponding PEHE (CATE error) and policy value can be estimated (e.g., using doubly robust estimators). Now, a $\gamma$ can be selected similar to an Elbow plot (e.g., for selecting the optimal number of clusters in a clustering algorithm): choose a value that results in a large policy value improvement, while resulting in a sufficiently accurate CATE estimator. For example, consider our real-world experiments (Fig. 6 in Sec. 5). Here, a reasonable value would be $\gamma= 0.8$, which is desirable in terms of both policy value improvement and CATE accuracy.
>
>     - If we primarily care about decision-making, we suggest using a large value, such as $\gamma=0.99$. This ensures that the resulting policy is comparable to standard off-policy learning (in terms of policy value), while still allowing for CATE interpretation. In contrast, the decision function for off-policy learning (i.e., $\gamma= 1$) has no interpretation as a treatment effect. The exact value (e.g., $\gamma \in \{0.98, 0.99\}$) can be determined via the same approach as in bullet point A.
>
> **Action:** We will discuss methods for selecting $\gamma$ more explicitly in the paper.
>
>
> ## Response to Questions
>
> 1. Thank you for this interesting question. In general, the discrepancy between CATE-based and optimal policy value is determined by how well the CATE projection using the model class $\mathcal{G}$ can estimate the sign of the ground-truth CATE $\tau^\ast$. This depends on several factors, including (i) the error of the optimal projection (related to model complexity), and (ii) the structure of the ground-truth CATE $\tau$.
>
>     As an illustration, consider Figure 2 (left) from our paper, where the discrepancy in policy value is characterized by the distance of the two points where the blue (estimator) and red line (ground-truth CATE) intersect zero. For (i), we could make the red line steeper (increasing the projection error), and thus can arbitrarily widen this gap to obtain an arbitrary discrepancy between CATE-based and optimal policy value. For (ii), we could simply shift the red line upwards, which would also widen the gap without changing the projection error.
>
>
>     If the projection error is small (i.e., our learned CATE is close to the ground-truth CATE), we are able to bound the discrepancy in policy value as follows:
>
>     **Lemma.** Let let the true CATE be $\tau(x)=\mu_{1}(x)-\mu_{0}(x)\in L^{2}(\mathbb{P})$. For a function class $\mathcal{G}\subset L^{2}(\mathbb{P})$, define its $L^{2}$‐projection of $\tau$ as $g^{\ast} = \arg\min_{g\in\mathcal{G}} \mathbb{E} \left[(\tau(X)-g(X))^{2}\right]$ and $\epsilon_2 =\|\tau-g^{\ast}\|_{L^2}$.
>
>     Furthermore, we introduce the corresponding thresholded treatment policies $\pi_{\tau}(x)=1(\tau(x)>0)$, $\pi_{g^{\ast}}(x)=\mathbf{1}(g^{\star}(x)>0)$ and the (shifted) policy value $V(\pi) = \mathbb{E} \bigl[\pi(X) \tau(X)\bigr]$. Then, it holds that $0 \leq V \bigl(\pi_{\tau}\bigr)-V \bigl(\pi_{g^{\star}}\bigr) \leq \epsilon_{2}$.
>
>     **Proof (sketch).** We define the *disagreement set* $\mathcal{M}:=\{ x | sign(\tau(x)) \neq sign(g^{\star}(x)) \}$. Then, it holds that
> $\Delta =V(\pi_{\tau})-V(\pi_{g^{\star}}) =\mathbb{E}\bigl[\tau(X)\bigl(\pi_{\tau}(X)-\pi_{g^{\star}}(X)\bigr)\bigr]
> =\mathbb{E}\bigl[\tau(X)\mathbf{1}_{\mathcal{M}}(X)\bigr]$. On $\mathcal{M}$ we have $\tau(X)g^{\star}(X)\le 0$, hence
> $|\tau(X)|\le|\tau(X)-g^{\star}(X)|$.
>
> This implies $\Delta \leq \mathbb{E}\bigl[|\tau(X)-g^{\star}(X)|1_{\mathcal{M}}(X)\bigr] \leq \mathbb{E}\bigl[|\tau(X)-g^{\star}(X)|\bigr] =|\tau-g^{\star}|_{1} \leq \epsilon_2$., where the last inequality follows from Hölder’s inequality.
>
> Note, however, that in many applications the projection error will be large as a result of constraints on the function class $\mathcal{G}$ to ensure, e.g., interpretability [1] or fairness [2, 3], or to accommodate small data regimes. In such scenarios, the discrepancy in policy values may be large, and our method would provide substantial benefits over classical two-stage learners.
>     **Action:** We will add the new theoretical result to our paper along with the above discussion.
>
> ## References
>
> [1] Tschernutter et al. Interpretable Off-Policy Learning via Hyperbox Search. ICML 2022.
>
> [2] Fang et al. (2021). Fairness-Oriented Learning for Optimal Individualized Treatment Rules. Journal of the American Statistical Association.
>
> [3] Kim et al. Fair and Robust Estimation of Heterogeneous Treatment Effects for Policy Learning. ICML 2023.

---

> > ### Comment · Reviewer_Yh69 · 2025-08-05
> >
> > Thanks for your response. I will maintain the score.

---

> > > ### Author Response · Authors · 2025-08-05
> > >
> > > Thank you! We sincerely appreciate your time and effort in reviewing our work. We will make sure to incorporate your points into the camera-ready version if case of acceptance.

---

### Note · Authors · 2025-08-13

We thank all reviewers for their time and thoughtful evaluations. We are encouraged that four out of five reviewers found our contribution novel and relevant and recommended acceptance.

We are grateful for the constructive feedback and will incorporate all points that came up during the rebuttal into the camera-ready version of our paper.  In particular:

- **Additional experiments and theory.** We will integrate the materials provided during the rebuttal: (1) experimental results with multivariate confounders and additional first-stage models (SNet and FlexTNet); and (2) theoretical results on the interplay among approximation error, the second-stage model, and the resulting discrepancy in policy value, along with further clarifications of the terms in Theorem 4.4.

- **Clearer scope and motivation.** We will sharpen the abstract and introduction to state that the paper addresses a fundamental question about the decision-making performance of **two-stage learners** under **constrained** second-stage models. We will add concrete scenarios where such constraints are crucial in practice (e.g., fairness, privacy, explainability, and small-sample regimes) and provide additional guidance on the selection of $\gamma$.

- **First vs. second stage: conceptual distinctions.** We will more clearly define first- and two-stage learners. In particular, we will explain **why RMNet cannot be written as a two-stage learner**: it does not estimate the CATE via direct minimization over a function class $\mathcal{G}$ but via the plugin approach $\tau(x) = \mu_1(x) - \mu_0(x)$. We will also clarify that a direct empirical comparison between first- and two-stage learners is outside our scope, as first-stage approaches do not permit imposing direct constraints on the CATE in the way our framework requires.

We believe these changes address the all of the reviewers’ concerns and further strengthen the paper. Thank you again, we sincerely appreciate your time and effort in reviewing our work and believe the revised paper will be a strong fit for NeurIPS 2025.

---

### Decision · Program_Chairs · 2025-09-17

**Decision:**

Accept (poster)

**Comment:**

In this paper, the authors propose an approach for optimal decision-making using two-stage CATE estimators. They demonstrated that a direct application of optimal CATE estimator can lead to suboptimal decision-making. To address this, they introduced a new objective that balances the CATE error and decision performance. The authors provided solid theoretical justification and empirical support for their claims.

The paper was well-received by most reviewers. Most reviewers (Reviewers Yh69, XmcU, x8UB, ytEC) acknowledged the importance of the problem, and several praised the paper's novelty (Reviewers Yh69, 9gJM, XmcU) and meaningful theoretical results (Yh69, 9gJM, XmcU, ytEC). In their rebuttal, the authors successfully addressed some key concerns, including the selection of $\gamma$ (raised by yh69, 9gJM, XmcU, ytEC) and comparisons with direct policy learning (raised by 9gJM, x8UB), which led to some reviewers raising their scores.

The only remaining negative rating ("borderline reject") came from Reviewer x8UB despite that the reviewer had increased the rating after the rebuttal. In my final discussion with the reviewer, Reviewer x8UB indicated that the only remaining concern is that the authors do not clearly state their focus on settings with limited model complexity in the title and abstract, as well as how it contrasts with the method without such restriction. However, Reviewer x8UB also acknowledged the paper's novelty within this setting. In my opinion, the focus on settings with limited complexity is still practically relevant, given the authors' examples of interpretability and fairness constraints listed in the response.

Overall, the paper received strong feedback, and its contributions outweigh the remaining negative criticism. Hence, I recommend acceptance.